# StepWiser: Stepwise Generative Judges for Wiser Reasoning

## Abstract

As models increasingly leverage multi-step reasoning strategies to solve complex problems, supervising the logical validity of these intermediate steps has become a critical research challenge. Process reward models address this by providing step-by-step feedback, but current approaches have two major drawbacks: they typically function as classifiers without providing explanations, and their reliance on supervised fine-tuning with static datasets limits generalization. Inspired by recent advances, we reframe stepwise reward modeling from a classification task to a reasoning task itself. We thus propose a generative judge that *reasons about the policy model's reasoning steps* (i.e., meta-reasons), outputting thinking tokens before delivering a final verdict. Our model, STEPWISER, is trained by reinforcement learning using relative outcomes of rollouts. We show it provides (i) better judgment accuracy on intermediate steps than existing methods; (ii) can be used to improve the policy model at training time; and (iii) improves inference-time search.

## 1 Introduction

As large language models (LLMs) increasingly tackle complex problems, they rely on multi-step reasoning strategies like Chain-of-Thought (CoT) (Wei et al., 2022) and ReAct (Yao et al., 2022) to decompose tasks and formulate better solutions. Consequently, ensuring these intermediate reasoning steps possess logical validity has become a critical research challenge. Process Reward Models (PRMs) have emerged as a potential tool to meet this need, providing step-by-step feedback for supervising learning, instead of relying on a single, often sparse, outcome-based reward (Lightman et al., 2023; Wang et al., 2023). However, this approach suffers from two major drawbacks. First, current PRMs typically function as "black-box" classifiers, providing a score or label without explaining why a step is correct or flawed. Second, their reliance on supervised fine-tuning (SFT) with static datasets can limit their ability to generalize to new reasoning patterns (Lightman et al., 2023; Luo et al., 2024; Wang et al., 2023; Xiong et al., 2024b; Zhang et al., 2024a). In contrast, reasoning models themselves are trained to produce CoTs with reinforcement learning (RL) for best performance (DeepSeek-AI et al., 2025).

In this paper we propose to reward intermediate reasoning steps by first **reasoning about those reasoning steps**, before making a judgment – a meta-reasoning process which itself is trained by RL. Our overall method (as shown in Figure 1) to build such a *stepwise generative judge* involves 3 components: (1) a new self-segmentation technique to equip the base policy model with the ability to produce coherent and informative reasoning chunks (chunks-of-thought); (2) assignment of target rewards to chunks via relative outcomes of rollouts; and (3) online training of judgment reasoning chains (i.e., reasoning about reasoning) and final reward judgments via RL. Our stepwise judge, termed STEPWISER, can then be used to provide rewards either at training time or inference time in order to improve the reasoning ability of the policy model.

We conduct a comprehensive evaluation of our method across three key dimensions: (i) the judge's classification accuracy on intermediate steps, e.g., via its score on ProcessBench (Zheng et al., 2024); (ii) its performance in a new inference-time search paradigm where the judge cleans up the reasoning history and re-samples – a method we propose for efficiently scaling sequential computation while maintaining the original generation length; and (iii) its utility in data selection for downstream model training. Our experiments demonstrate that our RL-trained generative stepwise judge significantly

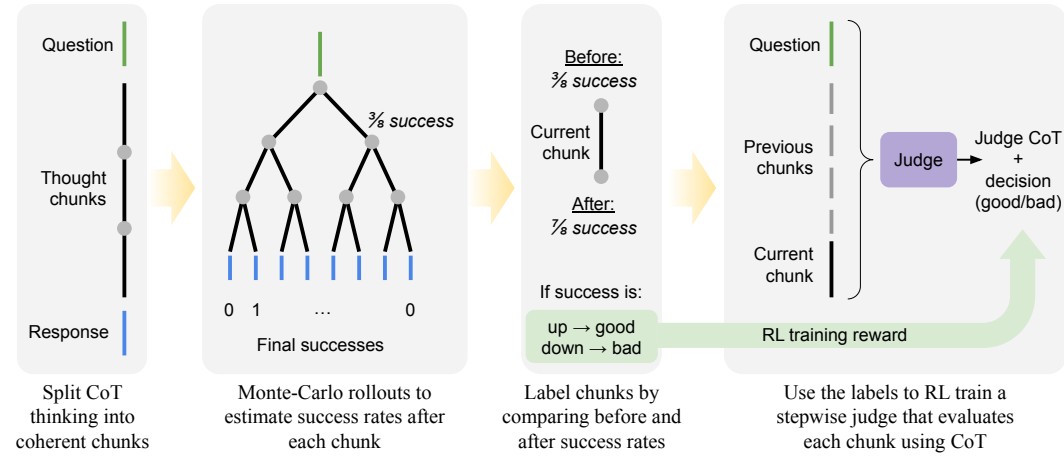

Figure 1: **Overview of our STEPWISER training method:** we teach the model to segment its chain-of-thought (CoT) into coherent chunks. Then after each chunk, we generate Monte-Carlo rollouts to estimate the average success rate (i.e. Q-value) starting from that point. If the success rate goes up (or down) after a given chunk, we label it as good (or bad). Using these labels, we RL train a stepwise judge model that determines the quality of a given chunk after its own CoT reasoning.

outperforms traditional SFT-based baselines and other existing methods across all axes of evaluation, where the ability to meta-reason – trained via RL – is the critical factor.

## 2 RELATED WORK

### 2.1 PROCESS REWARD MODELS IN LLM MATH REASONING

To improve the reliability of multi-step reasoning in LLMs, one can consider methods beyond evaluating only the final answer, termed Outcome Reward Models (ORMs), by instead evaluating each intermediate step, a method pioneered by Process Reward Models (PRMs). Lightman et al. (2023) first demonstrated that a process-supervised model can significantly outperform an outcome-supervised one in guiding best-of-n sampling. However, their PRM800K dataset relied on intensive human annotation for each reasoning step, which is generally infeasible for larger, more diverse and challenging datasets.

Subsequent research has focused on automating this annotation process. Wang et al. (2023) proposed using Monte Carlo (MC) rollouts to estimate the Q-value of each step, while Luo et al. (2024) introduces a binary search method to efficiently identify faulty steps. Our work builds upon the MC-based annotation approach, exploring various methods for converting these Q-value estimates into effective learning signals.

In parallel, another line of work has established a theoretical connection between intermediate step values and the final outcome within the framework of KL-regularized Markov Decision Processes (Zhong et al., 2024; Rafailov et al., 2024). This result has been used to derive DPO-like objectives for learning an implicit PRM from outcome-only data (Xiong et al., 2024a; Cui et al., 2025; Zhou et al., 2025) or a KL-regularized version of the MC-based estimator (Zhang et al., 2024a). A recent work (Zha et al., 2025) prompts LLMs to evaluate each individual step before producing a final judgment, but critically, supervises only the evaluation of the final answer. In contrast, our work demonstrates that providing explicit supervision for the evaluation of each intermediate step is a far more effective strategy. Our experiments conclusively show that leveraging these rich, dense feedback signals leads to a significantly more powerful and accurate judge model.

Concurrent work by He et al. (2025) uses a prompting approach to segment thought process into coherent chunks similar to ours. However, their stepwise judge is based only on prompting techniques that leverages hints in CoT like "Wait, I made a mistake". In contrast, our method focuses on training a judge using stepwise labels grounded in final verified answers.

## 2.2 Judge architectures

The process labels and signals described above can be used to train judges with different distinct architectures and training paradigms.

**Discriminative PRMs** The most straightforward approach is to treat the task as a *classification problem*. This involves replacing the language model's final layer with a linear head and fine-tuning it to predict a binary label for each step using a cross-entropy loss (Lightman et al., 2023). A more recent method formulates the task as next-token prediction, prompting the LLM to generate a pre-defined token (e.g., + or -) as its judgment (Wang et al., 2023; Xiong et al., 2024b). This approach further dates back to preference reward model training (Dong et al., 2024; Liu et al., 2023). Although this method uses a generative mechanism, its function remains purely discriminative, as it outputs a simple judgment without justification. We therefore group both under the discriminative category.

**Generative judges with CoT reasoning** In sharp contrast, the second and most recent paradigm is the generative reasoning judge. Here, the evaluation itself is framed as a reasoning task. The judge first generates an explicit CoT to explain its rationale before outputting its final judgment. This approach was initially explored for preference learning and ORMs (Zhang et al., 2024b; Chen et al., 2025). There are also a few very recent works studying this paradigm shift in the context of stepwise judges, including Zhao et al. (2025); Zha et al. (2025); Khalifa et al. (2025). Though we share similar spirit of leveraging the inherent reasoning ability of the LLMs to train a stepwise judge, the algorithmic designs are distinctly different.

**Comparison to Recent Work** First, in contrast to works focusing on offline rejection sampling fine-tuning (Zhao et al., 2025; Khalifa et al., 2025), we identified critical scalability issues with such static methods. While offline fine-tuning provides dramatic initial performance gains, we observed that learning quickly plateaus. Specifically, the model's loss stagnates after training on a relatively small dataset (e.g., 10k samples), preventing further improvement. In contrast, we observe that online RL framework allows for continuous learning and scalability, successfully showing sustained improvement on over 800k samples.

Second, our approach fundamentally diverges from RL methods that rely on sparse, trajectory-level supervision, such as Zha et al. (2025). Specifically, they prompt the LLMs to evaluate each individual step and final answer but only the final verification is supervised. Their approach assumes that to get an accurate evaluation of the final answer, models *implicitly become a stepwise judge*. Our divergence is rooted in strong theoretical evidence from preference learning, where an exponential gap in sample complexity can exist between learning from sparse final outcomes versus dense intermediate rewards (Zhong et al., 2024, c.f. Proposition 3.2).

This motivated our central hypothesis that a similar principle governs the training of generative judges. However, applying RL to decoupled, individual steps is non-trivial. We discovered that this fine-grained approach introduces unique challenges, namely (1) a susceptibility to majority class bias, and (2) noisy signals arising from naive trajectory segmentation. To address these, our framework incorporates two targeted solutions: self-segmentation fine-tuning to generate coherent reasoning steps, and prompt set balancing to mitigate bias. As our experiments will demonstrate, it is this complete framework—harnessing the power of explicit, stepwise signals while actively correcting for their inherent challenges—that is essential for training state-of-the-art generative judges.

## 3 Method: Training stepwise generative judges with RL

As depicted in Figure 1, our overall method STEPWISER consists of three components:

1. We equip the base policy model with the ability to self-segment Chain-of-Thoughts into coherent and informative reasoning chunks, called Chunks-of-Thought. This is done by creating SFT data with informative segments, so that the model can be trained to self-segment. We show that this causes no loss in performance for the base model and is critical to reduce the training noises during RL by removing the meaningless chunk.

2. Given the chunks generated by the policy model, we annotate each chunk to create training data for our generative stepwise judge with binary target labels. This is done by comparing outcomes of rollouts starting before and after the given chunk using the outcome rewards.

3. We perform online RL training using GRPO which trains our stepwise judge model to produce judgment reasoning chains (i.e., reasoning about reasoning) and reward final judgments that match the chunk labels from the previous step.

We describe the three components in detail in the following three subsections.

### 3.1 CoT Generation with Self-Segmentation (Chunks-of-Thought)

A core challenge in evaluating reasoning processes is defining what a "step" is. Simple heuristics, like splitting on double line breaks or predefined tokens like "Step 1, Step 2", often create fragmented or logically incomplete steps, making them difficult for a judge to evaluate. We present representative examples in appendix (Table 11-13), where the model tends to insert double line breaks before and after a mathematical equation. This breaks an intuitively unified logical step into multiple different chunks, where one chunk contains a textual explanation, and the next with the corresponding equation.

**Achieving better step definition via self-segmentation.** To mitigate this issue, we propose a method to teach the model to generate and simultaneously self-segment its own reasoning chains into more meaningful steps. First, we define the criteria for a high-quality reasoning step. The core idea is that each step should represent a complete logical leap or a self-contained part of the problem-solving process. Our definitions are given in Table 5. We then create our training data by:

1. Generating a set of initial reasoning trajectories from the base model.
2. Using an LLM prompted with our rules, to automatically segment these trajectories into logically coherent steps.

We fine-tune our base model on this data, thus teaching it to generate and simultaneously self-segment its own reasoning chains automatically. This self-segmentation ability is crucial for two main reasons. First, it produces more informative and logically complete steps, which provides better context for our judge model and improves its evaluation accuracy. Second, this method significantly reduces the total number of steps per trajectory. This reduction is also important because, as we will show, the process of annotating each step with a quality label is computationally expensive.

### 3.2 Stepwise Data Annotation

**Stepwise data annotation via Q value estimation**  Previous work has used human labelers to annotate correctness of each reasoning step (Lightman et al., 2023), although most such data is collected for proprietary models that we cannot access. Other works annotate steps automatically using methods like Monte Carlo estimation (Wang et al., 2023). We follow this second approach, using an estimated Q-value to measure the quality of each step.

For a given training prompt $x$ with verifiable outcome rewards, we generate a response from our policy model $\pi$ which segments its CoT into chunks $a = [a_1, a_2, \cdots, a_H]$, where $a_i$ is the i-th reasoning chunk. Then, the Q value of an individual step $a_i$ and its history is the expected final reward starting from that point:

$$Q^\pi\big([x, a_{1:i-1}], a_i\big) := Q^\pi(s_{i-1}, a_i) = \mathbb{E}_{a_{i+1:H} \sim \pi(\cdot|x, a_{1:i})} r^\star(x, a_{1:H}), \qquad (1)$$

where $s_i := [x, a_{1:i-1}]$ is the history, and $r^\star$ is a final reward, which can be 1 for correct answers and 0 otherwise. We estimate this Q-value by generating $M$ full completions $a_{i+1:H}^j$ from that step $a_i$ and calculating the average final reward, i.e. the ratio of correct final answers:

$$\widehat{Q}^\pi\big(s_{i-1}, a_i\big) = \frac{1}{M} \sum_{j=1}^{M} r^\star(x, a_{1:i}, a_{i+1:H}^j). \qquad (2)$$

Following prior work (Wang et al., 2023; Xiong et al., 2024b), we can then assign a binary label to the step based on this Q-value:

$$y_i = \begin{cases} + & \text{if } \widehat{Q}^\pi\big(s_{i-1}, a_i\big) > 0, \\ - & \text{if } \widehat{Q}^\pi\big(s_{i-1}, a_i\big) = 0. \end{cases}$$

For convenience, we refer to this labeling approach as *Absolute Q value thresholding* (`Abs-Q`).

Table 1: Prompt Template for our STEPWISER judge.

---

**Prompt Template for STEPWISER Judge**

**Instruction:**
You are a reasoning validator for mathematical problems. Your task is to think step by step and determine if the "New Reasoning Chunk" contains **any explicit errors** based on the problem description and historical context.

First, you must **always** perform a step-by-step chain of thought analysis to justify your final judgment. Then, based on your analysis, you will make a definitive judgment. It is OK that the chunk does not contain any numerical calculation.

Based on your evaluation, provide your final judgment:

- Use **Positive** if the reasoning chunk is free of mistakes.

- Use **Negative** if the reasoning chunk contains one or more mistakes.

---

Input:
Mathematical Problem: {problem}
Historical Reasoning Path: {history}
New Reasoning Chunk: {chunk}

---

**Output format:**

1. **Analysis**: [Always provide a step-by-step analysis here. First, briefly state the goal of the current reasoning chunk. Second, verify the logic, method, and any calculations against the problem's requirements and the historical path. If an error is found, clearly explain the error and why it's wrong. If the reasoning is correct, explain why it is a valid and logical step forward.]

2. **Final Judgment**: [Provide the final judgment within \boxed{}. Examples: \boxed{Positive} or \boxed{Negative}.]

---

**Rewarding the progress**   One drawback of `Abs-Q` is its insensitivity to the dynamics of the reasoning process. For instance, it does not differentiate between a step that raises the success probability from 10% to 50% and one that drops it from 60% to 55%. To reward progress, we also explore methods that consider the change in Q-value.

Setlur et al. (2024) proposes to consider the change in value. Specifically, they define the notion of *effective reward* as a combination of Q value and advantage function of the best-of-n policy induced by $r^\star$:

$$Q^\pi(s_{i-1}, a_i) + \alpha \cdot A^\mu(s_{i-1}, a_i), \qquad (3)$$

where $\alpha > 0$ is a hyperparameter, and $A^\mu(s_{i-1}, a_i) := Q^\mu(s_{i-1}, a_i) - Q^\mu(s_{i-2}, a_{i-1})$. Here $\mu$ is taken as the best-of-n policy with $r^\star$. In other words, we generate $n$ responses from $\pi$ and use $r^\star$ to select the best one. In this case, $\mu$ satisfies that $Q^\mu(s_{i-1}, a_i) = 1 - (1 - Q^\pi(s_{i-1}, a_i))^n$[1]. Therefore, the effective reward can also be estimated via Q value estimation. Accordingly, we consider an alternative approach of data annotation:

$$y_i = \begin{cases} + & \text{if } \widehat{Q}^\pi(s_{i-1}, a_i) + \alpha \cdot \widehat{A}^\mu(s_{i-1}, a_i) > 0, \\ - & \text{if } \widehat{Q}^\pi(s_{i-1}, a_i) + \alpha \cdot \widehat{A}^\mu(s_{i-1}, a_i) = 0, \end{cases}$$

where $\widehat{A}^\mu$ is the estimated advantage through the Monte-Carlo estimation of the Q value. We refer to this labeling approach as *Relative Effective Reward Thresholding* (`Rel-Effective`).

As a simpler alternative to capture relative improvement, we also consider a method based on the value ratio, where the label is determined by if $\widehat{Q}^\pi(s_{i-1}, a_i)/\widehat{Q}^\pi(s_{i-2}, a_{i-1}) > \gamma$. We refer this labeling approach as `Rel-Ratio`.

Using one of these methods, we can assign binary label $y_i$ to every step $a_i$ in a reasoning trajectory. Since these labels come from unbiased estimates of the actual Q-values, they are likely to be more reliable compared to more ad-hoc methods. For example, if a step $a_i$ is the first step with a mistake, rollouts starting after $a_i$ are more likely to fail compared to ones that start before the flawed step $a_i$.

---

[1]Assuming binary $\{0, 1\}$ outcome rewards where $(1 - Q^\pi(s_{i-1}, a_i))^n$ is the probability of $n$ rollouts failing.

### 3.3 TRAINING THE JUDGE VIA REINFORCEMENT LEARNING

Our goal is to train a stepwise judge using the segmented (chunked) reasoning chains and stepwise target labels. While a standard approach is to train a discriminative judge via SFT (Wang et al., 2023; Xiong et al., 2024b), we adopt a generative formulation inspired by recent studies (Zhang et al., 2024b; Chen et al., 2025; Whitehouse et al., 2025). We frame the evaluation as a reasoning task where the judge first generates a CoT analysis of the step in question, then outputs a final judgment. This generative process is compelling because it forces the judge to "show its work," leading to a more transparent and potentially more accurate evaluation.

**The Insufficiency of Offline Fine-tuning.** We first considered a simpler offline approach, rejection sampling fine-tuning (RFT), but found it has a critical limitation. On a static dataset, the training loss plateaus very quickly after very few optimization steps (Figure 2). This demonstrates that static methods are insufficient for this complex task, motivating our adoption of an online RL framework to leverage a continuous and more diverse set of learning signals.

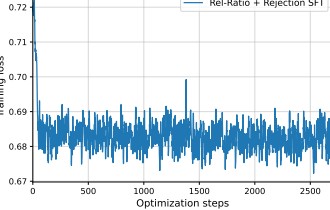

Figure 2: The training loss of offline RFT on a static dataset. The loss saturates quickly, indicating that learning has stagnated.

**RL Task Formulation and Training.** We formulate the training as a step-level judgment task. For each step, the judge model is given the original problem $x$, the reasoning history $a_{1:i-1}$, and the current chunk $a_i$ to evaluate. It is then prompted to generate a CoT rationale explaining its analysis, followed by a final verdict (see prompt template in Table 1).

**Prompt balancing.** A critical challenge we identified in this stepwise setting is the severe label imbalance produced by our data annotation process (e.g., over 70% positive labels with `Abs-Q`). Our early experiments showed this leads to degenerate judges that simply learn to always predict majority class. To address this, we incorporate **prompt dataset balancing** by down-sampling the majority class to ensure a 1:1 ratio of positive and negative examples. This simple technique proved essential for stable training and robust performance, an impact we quantify in our ablation study in Section 4.3.

The RL training itself is straightforward. The judge receives a reward of 1 if its verdict matches the target label $y_i$ and 0 otherwise. We use the GRPO algorithm (Shao et al., 2024) for optimization.

## 4 EXPERIMENTS

We use the Qwen2.5-1.5B-it and 7B-it models (Yang et al., 2024) as our base models. The prompts come from a subset of NuminaMath-CoT dataset (Beeching et al., 2024), which we preprocess by removing duplicates and filtering for problems verifiable by Math-Verify. This section highlights a key result that enables our pipeline's feasibility; full implementation details for all training stages, hyperparameters, and additional ablations are deferred to Appendix B.

### 4.1 SELF-SEGMENTATION FINE-TUNING

We first fine-tune the base policy to segment its own reasoning into meaningful "chunks". This is achieved by generating solutions to 20k problems, using a powerful teacher model (Llama-3.1-70B-it) to segment them into high-quality steps, and then fine-tuning our policy on this data. As shown in Table 6, this crucial pre-training step significantly reduces the number of steps per solution (e.g., from 9.6 to 6.0) without harming task performance.

This reduction in complexity is vital for two reasons: it (1) substantially lowers the computational cost of our subsequent data annotation stage, and (2) provides a cleaner, less noisy signal for the final RL training by filtering out trivial or meaningless steps. We provide detailed ablation studies in Appendix C.1 that further verify the effectiveness of this approach.

Table 2: **ProcessBench results.** Average accuracy (Avg) of our method STEPWISER is better than all variants of our discriminative baselines, and existing baselines in the literature (first rows). Further comparisons are given in Appendix Table 9.

| Method | Learning signal | GSM8K | MATH | Olympiad | Omni-MATH | Avg ↑ |
|---|---|---|---|---|---|---|
| *Existing Reference Models* | | | | | | |
| Math-Shepherd-PRM-7B | `Abs-Q` | 47.9 | 29.5 | 24.8 | 23.8 | 31.5 |
| RLHFlow-Llama3-8B-it | `Abs-Q` | 50.4 | 33.4 | 13.8 | 15.8 | 28.4 |
| Skywork-Qwen2.5-Math-7B-it | `Abs-Q` | 70.8 | 53.6 | 22.9 | 21.0 | 42.1 |
| Eurus-Qwen2.5-Math-7B-it (DPO) | Outcome | 56.6 | 43.0 | 27.3 | 26.8 | 35.1 |
| RL-TANGO-Qwen2.5-7B-it | Outcome | 53.1 | 48.2 | 37.8 | 36.3 | 43.9 |
| *Qwen2.5-1.5B-chunk* | | | | | | |
| Discriminative + SFT | `Abs-Q` | 39.3 | 32.1 | 19.3 | 18.9 | 27.2 |
| Discriminative + SFT | `Rel-Effective` | 40.8 | 37.2 | 18.7 | 20.1 | 29.2 |
| Discriminative + SFT | `Rel-Ratio` | 32.1 | 32.0 | 14.2 | 18.0 | 24.1 |
| Generative CoT + RL (STEPWISER) | `Abs-Q` | 49.2 | 40.5 | 23.8 | 31.0 | 36.1 |
| Generative CoT + RL (STEPWISER) | `Rel-Effective` | 48.2 | 43.6 | 22.1 | 25.3 | 34.8 |
| Generative CoT + RL (STEPWISER) | `Rel-Ratio` | 46.9 | 43.4 | 26.3 | 28.4 | 36.2 |
| *Qwen2.5-7B-chunk* | | | | | | |
| Discriminative + SFT | `Abs-Q` | 54.8 | 45.9 | 28.0 | 26.9 | 38.9 |
| Discriminative + SFT | `Rel-Effective` | 55.6 | 48.7 | 26.4 | 28.3 | 39.7 |
| Discriminative + SFT | `Rel-Ratio` | 48.6 | 46.9 | 21.9 | 25.4 | 35.7 |
| Generative CoT + RL (STEPWISER) | `Abs-Q` | 61.9 | 61.0 | 48.4 | 43.9 | 53.8 |
| + Maj@8 | `Abs-Q` | 65.5 | 62.1 | 49.7 | 45.7 | 55.8 (+2.0) |
| Generative CoT + RL (STEPWISER) | `Rel-Effective` | 72.4 | 68.3 | 54.4 | 52.4 | 61.9 |
| + Maj@8 | `Rel-Effective` | **72.9** | **72.1** | **57.3** | **54.0** | **64.1 (+2.2)** |
| Generative CoT + RL (STEPWISER) | `Rel-Ratio` | 72.6 | 67.2 | 52.3 | 49.8 | 60.5 |
| + Maj@8 | `Rel-Ratio` | 74.3 | 69.0 | 53.8 | 50.2 | 61.8 (+1.3) |

## 4.2 EVALUATION ON PROCESSBENCH

We first evaluate STEPWISER on ProcessBench (Zheng et al., 2024), a benchmark for identifying the first incorrect step in a reasoning process. Performance is measured by the harmonic mean of accuracy on problems with correct ($\mathrm{acc}_1$) and incorrect ($\mathrm{acc}_2$) final answers, calculated as $2 \times (\mathrm{acc}_1 \times \mathrm{acc}_2)/(\mathrm{acc}_1 + \mathrm{acc}_2)$.

**STEPWISER significantly outperforms SFT and RL baselines.** Our primary results in Table 2 show that our RL-trained generative judge, STEPWISER, consistently and substantially outperforms baselines. First, it is far superior to SFT-trained discriminative judges across all signals and model scales. For instance, our 7B model with the `Rel-Effective` signal scores 61.9, beating the SFT baseline's 39.7. Its performance also surpasses that of similar community-trained PRMs.

**Our RL-trained STEPWISER judge significantly outperforms existing RL-trained judges.** Furthermore, we benchmark STEPWISER against other models trained with online methods like online DPO (Xiong et al., 2023; Xu et al., 2023) or GRPO (Shao et al., 2024) (e.g., Eurus-7B, RL-TANGO-7B). Unlike our method, these models are supervised at the *trajectory level*, using only the final answer's correctness as a reward signal, denoted by "Outcome" in Table 2. For instance, our best 7B model's score of 61.9 is well ahead of RL-TANGO's 43.9. This result provides strong evidence for our core hypothesis: that direct, dense, step-level supervision provides a much richer and more effective learning signal for training process reward models.

## 4.3 ANALYSIS OF KEY COMPONENTS TO THE PERFORMANCE GAP

To understand the source of this performance gap, we conduct a series of ablation studies comparing our full STEPWISER method against baselines that remove one key component at a time: *(1) Ablate RL* by using offline rejection sampling (RFT) (Dong et al., 2023); *(2) Ablate CoT* by training a discriminative judge with RL where the model directly outputs a token to indicate the judgment; and *(3) Ablate prompt dataset balancing*. For brevity, we focus on the `Rel-Ratio` signal here, as the trends are consistent across others (see Appendix for more).

**Online RL contributes to the performance improvement.** The importance of online learning is evident when comparing our full STEPWISER model to the RFT baseline. On ProcessBench using

Table 3: Ablation study results on ProcessBench. The results show that both the generative CoT reasoning and RL components of our STEPWISER method are important for overall results.

| Method | GSM8K | MATH | Olympiad | Omni-MATH | Avg ↑ |
|---|---|---|---|---|---|
| *Qwen2.5-1.5B-chunk* | | | | | |
| Discriminative + SFT (Baseline) | 32.1 | 32.0 | 14.2 | 18.0 | 24.1 |
| **STEPWISER (Generative Reasoning + RL)** | **46.9** | **43.4** | **26.3** | 28.4 | **36.2** |
| – Ablate RL (use RFT) | 32.8 | 23.9 | 16.3 | 19.6 | 23.1 |
| – Ablate CoT (use Discriminative format + RL) | 42.0 | 43.2 | 23.6 | **28.7** | 34.3 |
| *Qwen2.5-7B-chunk* | | | | | |
| Discriminative + SFT (Baseline) | 48.6 | 46.9 | 21.9 | 25.4 | 35.7 |
| **STEPWISER (Generative Reasoning + RL)** | **72.6** | **67.2** | **52.3** | **49.8** | **60.5** |
| – Ablate CoT (use Discriminative format + RL) | 58.7 | 49.4 | 40.8 | 42.7 | 47.9 |
| – Ablate Prompt Balancing (Generative Reasoning + RL) | 58.8 | 54.8 | 41.0 | 36.9 | 47.9 |

Qwen2.5-1.5B-chunk, the RFT model achieves an average score of only 23.1, which is substantially lower than STEPWISER's score of 36.2 and is even worse than the standard discriminative SFT baseline (24.1). From Figure 2, we notice that its training loss on a large, static dataset plateaus quickly. This trend is consistent across other learning signals and the larger 7B model, indicating that offline methods are insufficient to capture the complexity of CoT reasoning and reward modeling, making online RL a critical component.

**STEPWISER judge with CoT leverages intrinsic reasoning ability to obtain better evaluation** The benefit of the generative CoT format is illustrated by the "Ablate CoT" baseline. With the Qwen2.5-1.5B-chunk model, augmenting a discriminative-style judge with RL boosts the ProcessBench score from 24.1 (SFT) to 34.3 (RL), but it still falls short of the STEPWISER model's 36.2. Moreover, the in-distribution accuracy results in Figure 5 show that the STEPWISER model with CoT reasoning achieves higher accuracy on the held-out data. This suggests that generating explicit rationales provides a more expressive and informative structure for learning and modeling the stepwise reward signal. The gap between the generative CoT model and the discriminative model becomes much larger with the stronger Qwen2.5-7B-chunk. Specifically, the generative STEPWISER model reaches an average score of 60.5, while the discriminative model only achieves 47.9. This is because we are leveraging the intrinsic reasoning ability of the base model through CoT in the judgment so the stronger model offers more advantages.

**Prompt dataset balancing stabilizes training and mitigates overfitting.** The practice of balancing the prompt dataset is also crucial for robust performance. Our ablation study on the Qwen2.5-7B-chunk model shows that removing this balancing step causes a substantial performance drop, with the average ProcessBench score dropping from 60.5 to 47.9. A deeper analysis reveals that while both the "Ablate CoT" ablation and the lack of dataset balancing hurt performance, their underlying failure modes are different. The "Ablate CoT" model suffers from a general decline in its ability to recognize correct and incorrect steps. In contrast, without balancing, the prompt dataset is heavily biased towards positive examples. This trains the model to be overly optimistic, developing a strong bias towards predicting any given step as correct. This bias is particularly enhanced during online training, which eventually leads to training instability and model collapse. A detailed analysis of this phenomenon is provided in the Appendix C.2.

### 4.4 USING THE STEPWISER JUDGE TO OBTAIN BETTER SOLUTIONS

In this section, we evaluate the practical utility of our RL-trained STEPWISER judge by using it to guide an LLM's reasoning process at inference time. We employ a search strategy called *Chunk-Reset Reasoning*. The base policy model generates a solution "chunk-by-chunk". After each chunk is produced, our STEPWISER judge evaluates it. If the chunk is deemed correct, it is accepted, and the model proceeds. If it is rejected, the flawed chunk is discarded, and the policy model re-generates a new one from the same point (up to 5 attempts). This allows the model to explore alternative reasoning paths without committing to an early mistake. This paradigm effectively scales sequential compute (i.e., compute used to extend a single trajectory) while keeping the final accepted token count similar.

Table 4: **Inference time search via Chunk-Reset Reasoning.** We report results with both Qwen2.5-1.5B-chunk and Qwen2.5-7B-chunk, using them as both the response generators and the initialization checkpoints for the STEPWISER judge. We see clear improvements using STEPWISER across both model sizes, with similar accepted responses lengths (on MATH500). Rejected length is the number of tokens in removed chunks during inference time search.

| Method | Learning signal | MATH500 | NuminaMath Heldout-1K | Avg ↑ | Accepted length | Rejected length |
|---|---|---|---|---|---|---|
| *Qwen2.5-1.5B-chunk* | - | 44.7 | 17.6 | 31.2 | 616.0 | 0.0 |
| Discriminative + SFT | `Abs-Q` | 47.7 | 19.1 | 33.4 | 625.2 | 218.7 |
| Discriminative + SFT | `Rel-Effective` | 47.4 | 19.6 | 33.5 | 612.7 | 302.4 |
| Discriminative + SFT | `Rel-Ratio` | 50.4 | 20.0 | 35.2 | 596.0 | 475.8 |
| Generative CoT + RL (STEPWISER) | `Abs-Q` | 51.4 | 19.8 | 35.6 | 599.1 | 1069.2 |
| Generative CoT + RL (STEPWISER) | `Rel-Effective` | 52.1 | 21.2 | 36.7 | 602.0 | 947.4 |
| Generative CoT + RL (STEPWISER) | `Rel-Ratio` | 51.9 | 21.8 | 36.9 | 596.4 | 884.7 |
| *Qwen2.5-7B-chunk* | - | 73.3 | 41.5 | 57.4 | 609.5 | 0.0 |
| Discriminative + SFT | `Abs-Q` | 74.8 | 44.4 | 59.6 | 654.0 | 168.2 |
| Discriminative + SFT | `Rel-Effective` | 76.9 | 46.1 | 61.5 | 654.6 | 186.5 |
| Discriminative + SFT | `Rel-Ratio` | 76.7 | 45.8 | 61.3 | 641.4 | 219.7 |
| Generative CoT + RL (STEPWISER) | `Abs-Q` | 77.5 | 46.3 | 61.9 | 658.5 | 345.7 |
| Generative CoT + RL (STEPWISER) | `Rel-Effective` | 78.3 | **48.1** | 63.2 | 660.8 | 425.8 |
| Generative CoT + RL (STEPWISER) | `Rel-Ratio` | **79.0** | 47.5 | **63.3** | 653.0 | 295.4 |

**Inference-time search consistently improves performance.** As shown in Table 4, using our STEPWISER judge for guidance leads to superior outcomes. With the `Rel-Ratio` learning signal, our approach steers the 1.5B model to an average accuracy of 36.9%, a significant improvement over the 31.2% of the base model without guidance. We observe a clear trend of our STEPWISER judge being superior to the discriminative models across all learning signals, and this trend holds for the 7B model, demonstrating the scalability of our approach.

**Superior error detection enables effective self-correction.** The "Accepted Length" column in Table 4 shows that the final solutions are of a similar length to the baselines. However, the "Rejected Length" column, which tracks discarded chunks, is significantly higher when using STEPWISER. We interpret this as direct evidence of STEPWISER's superior ability to identify incorrect or unproductive steps. This triggers the reset mechanism more effectively, forcing the model to discard flawed reasoning and find a better path, which is consistent with its higher accuracy on ProcessBench.

**Relative signals prove more effective for guidance.** The inference-time search results also reinforce a key finding of this paper: training signals that reward relative progress (`Rel-Effective`, `Rel-Ratio`) consistently yield better judges than a signal that only measures a step's absolute quality (`Abs-Q`). For example, the `Rel-Effective` judge guides the 7B model to 64.3% accuracy, outperforming the `Abs-Q` judge (61.9%). This pattern, further confirmed by our data selection experiments in Appendix C.4, shows that modeling the dynamics of reasoning is a more effective strategy for training useful judges.

## 5 CONCLUSION

Reasoning models that output internal thought tokens before a final response have proven to outperform non-reasoning models. In this paper we have shown that further improvements can be found by making models ***reason about the reasoning decisions*** made within those internal thoughts. We provide a recipe to: (1) segment reasoning into chunks-of-thought; (2) assign rewards to chunks via relative outcomes of rollouts; and (3) train a judge model to reason about the quality of CoT chunks via reinforcement learning (RL).

Our stepwise generative judge STEPWISER is shown to be superior to existing methods on Process-Bench, to provide improved inference time search, and better training time rewards for building better response models. We show that both the use of reasoning during judgment, and training with RL in order to reason about reasoning, are important components to achieve this performance.

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

## A LLM USAGE STATEMENT

In the final stages of preparing this manuscript, we utilized a large language model (LLM) as a writing assistant. The scope of its use was strictly limited to proofreading and refining the grammar, and clarity of the text after all scientific work, experiments, and the initial draft were completed by the authors. All authors reviewed, edited, and approved the LLM's suggestions and take full responsibility for the final content of this paper.

Table 5: Rules that we provide for an LLM to create segmented Chunks-of-Thought SFT data.

**Rules for CoT Trajectory Segmentation**

**Segmentation Principles**

1. Unified purpose: A chunk should serve a single, clear objective. For example: setting up an initial equation, executing a self-contained calculation (like integration by parts), or stating a final/intermediate conclusion. All content within the chunk must directly serve this one core goal.

2. Logical Cohesion: All lines within a chunk must form a continuous and uninterrupted logical flow. A new chunk should begin as soon as the focus or purpose of the reasoning shifts.

3. Clear Transition: A new chunk must begin when the problem-solving process enters a new phase. This includes transitioning from "solving for a variable" to "verifying the answer," or inserting an "explanatory side-note" into the main workflow.

**Format rules.**

1. Use `<chunk>`... `</chunk>` to mark the beginning and end of each segment. The text and newlines inside the tags must not be altered.

2. The final output should only contain the tagged content, without any additional text, titles, or blank lines.

3. You **must** preserve all original text and newlines exactly as they appear within the tags.

Table 6: Comparison of the base policy with and without self-segmentation fine-tuning. Overall performance is comparable, but self-segmentation results in less chunks than using split by \n \n. Here Avg@32 is the test accuracy averaged over 32 trajectories with random seeds.

| Generator | Method | # Steps | # Tokens | Avg@32 on MATH500 |
|---|---|---|---|---|
| Qwen2.5-1.5B-it | Split by \n\n | 9.6 | 686.7 | 44.2 |
| Qwen2.5-1.5B-chunk | Self-segmentation | 6.0 | 714.1 | 44.7 |
| Qwen2.5-7B-it | Split by \n\n | 9.9 | 733.0 | 73.3 |
| Qwen2.5-7B-chunk | Self-segmentation | 6.8 | 768.1 | 73.3 |

## B EXPERIMENT SETUPS AND IMPLEMENTATION DETAILS

### B.1 BASE MODELS AND DATA PREPROCESSING.

The base models used in our experiments are `Qwen2.5-1.5B-it` and `Qwen2.5-7B-it` (Yang et al., 2024), both featuring a context window of 8192 tokens. Our training data is sourced from the NuminaMath-CoT dataset (Beeching et al., 2024). Ground-truth verification for mathematical problems is performed using the Math-Verify tool. The preprocessing pipeline for the training data is as follows:

- **Deduplication:** Duplicate prompts are removed from the dataset.
- **Verification and Filtering:** We use Math-Verify to extract and score the final answer from each reference solution. Prompts where the answer cannot be reliably verified are discarded.

Unless stated otherwise, the same base model is used to initialize both the policy and the judge.

### B.2 IMPLEMENTATION DETAILS OF SELF-SEGMENTATION FINE-TUNING

To enable our models to structure their own reasoning, we first established a set of principles for segmenting CoT trajectories into meaningful "chunks". These rules, detailed in Table 5, are designed to guide a powerful teacher model in creating a high-quality dataset for subsequent self-segmentation fine-tuning.

We then used these rules to generate the self-segmentation dataset via the following multi-step pipeline:

- **Initial Generation:** From a random subset of 20k NuminaMath-CoT prompts, we generate 16 responses per prompt using the base policy model.
- **Solution Filtering:** We discard incorrect responses, keeping a maximum of 4 correct solutions for each prompt.
- **Segmentation by Teacher Model:** We prompt a powerful teacher model, `Llama-3.1-70B-it` (Meta, 2024), to segment the correct solutions into meaningful chunks based on the rules outlined in Table 5.
- **Segmentation Filtering:** For each solution, we generate 8 segmented versions and retain only those that perfectly reconstruct the original text and adhere to the specified format.

The base model is then fine-tuned on this curated dataset. Fine-tuning is performed using the Axolotl package with the following hyperparameters: a learning rate of $1e-5$, a packing block size of 8192, and a global batch size of 32. The prompt template is provided in Table 10. This process successfully teaches the model to generate more structured reasoning, significantly reducing the number of steps compared to naive splitting methods, as detailed in Table 6. Meanwhile, we observe that for most current open-source thinking models that do long reasoning before answering, the number of steps exceeds 150 when trajectories are segmented using `\n\n`, with each step containing only about 30 tokens. While a broader application of our technique is beyond the scope of this work due to resource constraints, we believe it holds particular promise for these long-reasoning scenarios, which we leave for future exploration.

### B.3 Details of Stepwise Data Annotation

To automatically generate supervisory signals for each reasoning step, we adopt a framework based on Monte Carlo estimation of Q-values. In Section 3.2, we described our high-level approach for data annotation. Here, we provide the detailed mathematical formulations.

#### B.3.1 Stepwise data annotation via Q value estimation

#### B.3.2 Implementation Details of Stepwise Data Annotation

We select a subset of $40k$ prompts from NuminaMath for stepwise data annotation based on a pre-filtering process using the pass@k metric. Specifically, for each prompt, we generate 16 responses using our chunk-tuned models (e.g., Qwen2.5-1.5B-chunk). To ensure the selected prompts are of a suitable difficulty, we filter out prompts where the responses were either all correct or all incorrect. During generation, we use a temperature of 1.0 and set the maximum token limit to $8192$, or until the model produced a final answer. Then, for each intermediate step in a solution, we sample another $M = 16$ completions staring from that step for estimating Q-values.

While we follow the well-established annotation framework from prior literature (Wang et al., 2023; Xiong et al., 2024b), we note that advanced techniques like model ensembles or human verification could further enhance label quality (Zhang et al., 2025). These engineering improvements are orthogonal to our core investigation and could be integrated in future work. The full annotation process is rather computationally intensive, taking approximately 14 days on 8 A100 GPUs for the Qwen2.5-7B-chunk model. Notably, the self-segmentation fine-tuning described previously plays a crucial role here, as it significantly reduces the total number of chunks per trajectory, thereby saving substantial compute and annotation time.

### B.4 Judge Model Training Details

We train and compare two types of stepwise judges: a discriminative judge trained via SFT, which serves as a strong baseline, and our proposed generative judge trained with RL, which learns to produce CoT reasoning before its final decision.

#### B.4.1 Preliminaries: Hyperparameter Search for Labeling Signals

We conduct hyperparameter tuning for the learning signals labeling. We mainly search by training discriminative models and SFT training, as this is more computationally efficient than full RL training. For `Rel-Ratio`, we search over $\gamma \in \{0.6, 0.7, 0.8, 1.0, 1.2\}$, and for `Rel-Effective`,

we search over $\alpha \in \{0.2, 0.4, 0.6, 0.8, 1.0\}$ with $\mu$ set as the best-of-4 policy induced by the base policy. For Qwen2.5-1.5B-chunk, we choose $\gamma = 0.8$ and $\alpha = 0.4$, while for Qwen2.5-7B-chunk, we use $\gamma = 0.7$ and $\alpha = 0.8$.

### B.4.2 BASELINE: DISCRIMINATIVE JUDGE VIA SFT

We follow Xiong et al. (2024b) to formulate the discriminative stepwise judge as a multi-turn conversation task. Specifically, in every user turn, we provide a single step of reasoning, while in the next assistant turn, the model will decode either "+" or "-" token to indicate its judgment.

For training, we use standard SFT code. The data is packed into a block with length 8192 tokens. We use a learning rate of $1e-5$, a global batch size of 32. We also mask out the user turn's loss. We present the representative training loss curves in Figure 3.

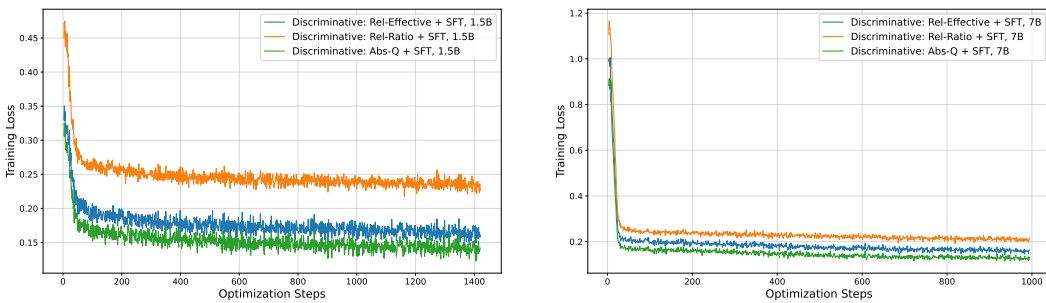

Figure 3: The training loss curves of discriminative stepwise judge under different learning signals. Left: 1.5B model, Right: 7B model.

### B.4.3 OUR METHOD: GENERATIVE JUDGE VIA RL

We train the judge model using the GRPO algorithm (Shao et al., 2024), implemented with the verl library (Sheng et al., 2024).

- **Hyperparameters:** We use a learning rate of $1e-6$, a per-prompt batch size of 1024, and a gradient update mini-batch size of 256. The judge generates 4 responses per prompt. The maximum prompt length is set to 3096 tokens, and the model can generate up to 3096 new tokens.

- **Training Stability:** We identified and addressed two primary stability issues. First, to counteract rapid entropy decay and the resulting zero-gradient problem, we employed the **clip higher** technique (Yu et al., 2025) with $\epsilon_h = 0.28$ and $\epsilon_l = 0.2$. Second, to mitigate bias from imbalanced labels, we created a **balanced training set** by down-sampling the majority class. We also apply a heuristic filtering process to remove prompts that were overly short or excessively long.

The RL training for the 7B model took approximately 5 days on 8 A100 GPUs. Figure 4 illustrates the training dynamics. The model is Qwen2.5-7B-chunk and the learning signal is `Rel-Ratio` with threshold 0.7. We can see that clip higher helps to encourage exploration and leads to a higher training curve.

## C ADDITIONAL RESULTS AND ABLATION STUDIES

### C.1 ABLATION ON SELF-SEGMENTATION

To validate the effectiveness of our self-segmentation approach, we compare its performance against a naive baseline that splits reasoning trajectories by \n\n. The results, presented in Table 7, show that the benefits of self-segmentation are most apparent in the context of RL training.

While the average scores for SFT models are comparable (e.g., 27.5 vs. 27.2 for `Abs-Q`), models trained with RL show significant improvements. Specifically, the average score for `Abs-Q + RL`

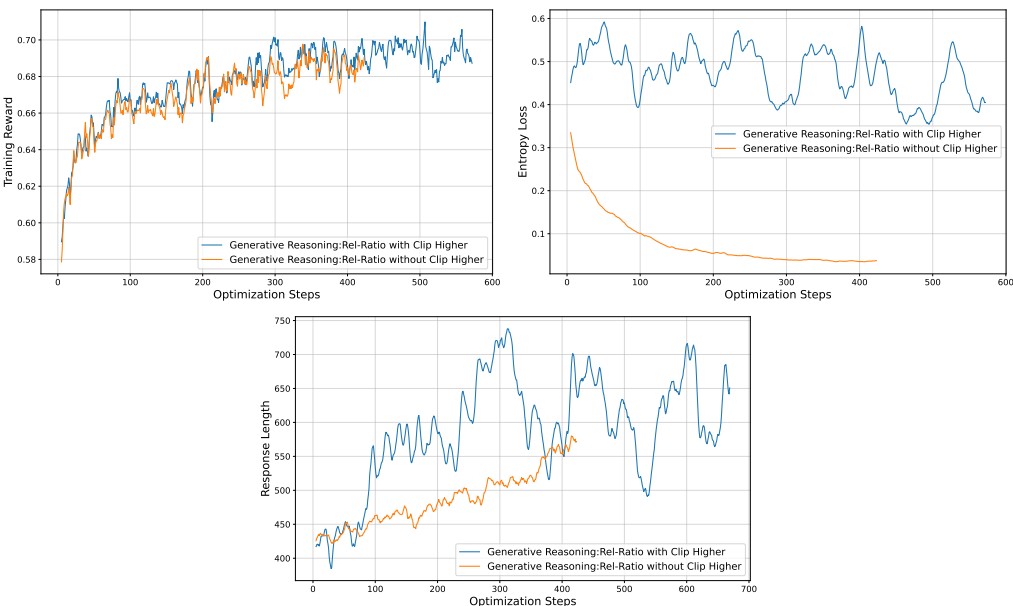

Figure 4: A representative example of training reward, entropy loss, and response length with and without clip higher technique. The model is Qwen2.5-7B-chunk and the learning signal is `Rel-Ratio` with threshold 0.7.

Table 7: The main ablation results on self-segmentation fine-tuning an chunking.

| Method | Learning signal | # Steps | GSM8K | MATH | Olympiad | Omni-MATH | Ave |
|---|---|---|---|---|---|---|---|
| Split by \n\n | `Abs-Q` + SFT | 5457820 | 33.7 | 37.1 | 20.2 | 18.9 | 27.5 |
| Split by \n\n | `Abs-Q` + RL | - | 46.3 | 38.4 | 19.0 | 25.8 | 32.4 |
| Split by \n\n | `Rel-Ratio` + SFT | - | 28.3 | 30.9 | 15.5 | 21.0 | 23.9 |
| Split by \n\n | `Rel-Ratio` + RL | - | 46.3 | 39.1 | 17.3 | 21.3 | 31.0 |
| Self-segmentation | `Abs-Q` + SFT | 3463520 | 39.3 | 32.1 | 19.3 | 18.9 | 27.2 |
| Self-segmentation | `Abs-Q` + RL | - | 49.2 | 40.5 | 23.8 | 31.0 | 36.1 |
| Self-segmentation | `Rel-Ratio` + SFT | - | 32.1 | 32.0 | 14.2 | 18.0 | 24.1 |
| Self-segmentation | `Rel-Ratio` + RL | - | 46.9 | 43.4 | 26.3 | 28.4 | 36.2 |

increased from 32.4 to 36.1, and for `Rel-Ratio` + RL from 31.0 to 36.2. This disparity suggests that the self-segmentation process produces a cleaner, more meaningful step-wise signal by filtering out noisy or trivial intermediate steps. In particular, we refer the interested readers to Table 11-13 for detailed examples. The RL process, being more sensitive to data and reward quality, benefits greatly from this higher-quality signal. Conversely, SFT appears more robust to this type of noise, and thus its performance is less impacted.

## C.2  Ablation on Key Components of the Generative Judge: CoT and Prompt Dataset Balancing

To understand the contributions of the core components of our generative judge, we conduct two key ablation studies presented in Table 9: (1) removing the generative CoT reasoning and (2) disabling the dataset balancing mechanism. The results reveal that while the absence of either component degrades the final F1 score, the underlying reasons for the performance drop are fundamentally different.

**The Impact of CoT Reasoning**   In this ablation, we train the judge to directly output a final judgment without generating any intermediate reasoning steps. We apply a format penalty of -1.0 if the model fails to follow this instruction. The results show that removing CoT weakens the model's

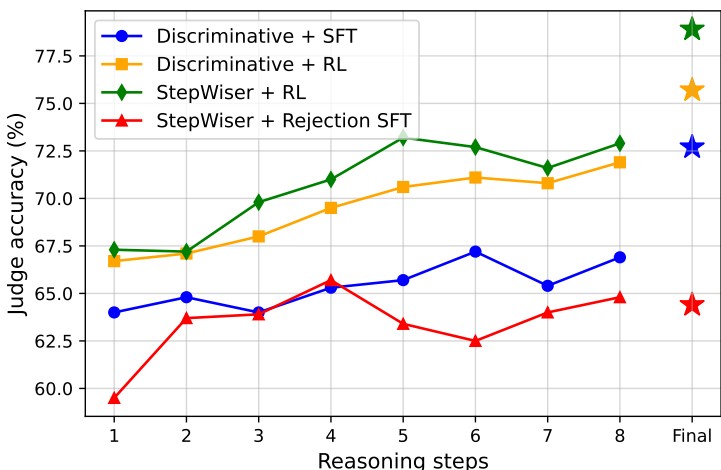

Figure 5: STEPWISER ablation results. **Left:** Test stepwise accuracy of various stepwise judge setups. Both generative CoT and RL training are important for the best stepwise judge. Here we plot the results of `Rel-Ratio` using Qwen2.5-1.5B-chunk, other results are presented in Figure 6.

overall ability to discriminate between correct and incorrect reasoning. This is evidenced by a general decline in accuracy for both "Correct" and "Error" classifications, suggesting that the act of generating a rationale is integral to the judge's reasoning capability.

**The Impact of Dataset Balancing**   We also examine the effect of training on the original, imbalanced data, where positive labels can be overrepresented (e.g., 70.2% for `Abs-Q`). Removing the balancing mechanism introduces a strong class bias. The model learns to over-predict the majority class ("Correct"), leading to a sharp increase in accuracy for correct steps but a catastrophic drop in its ability to detect errors. This trade-off is ultimately detrimental, as a judge that cannot identify mistakes is of little practical use, which is reflected in the significant F1 score degradation.

## C.3   ADDITIONAL RESULT ON CLASSIFICATION ACCURACY

Figure 5 and 6 compares the stepwise classification accuracy of our generative judge (STEPWISER) against the discriminative baseline across the different learning signals.

For the relative signals (`Rel-Ratio` and `Rel-Effective`), our RL-trained generative judge achieves significantly higher test accuracy on both intermediate steps and final answer evaluation. This suggests that the process of generating CoT reasoning provides the model with greater expressive capacity, enabling it to better capture these nuanced, dynamic signals.

In contrast, the performance gap narrows for the `Abs-Q` signal. We attribute this to a data distribution shift: the original `Abs-Q` dataset is highly imbalanced (70.2% positive samples), and the necessary downsampling to stabilize RL training adversely affects the judge's performance on the original, imbalanced test set. Nevertheless, even under these conditions, our generative judge remains substantially more accurate at the crucial task of verifying the final answer's correctness.

## C.4   APPLICATION: DATA SELECTION VIA REJECTION SAMPLING FINE-TUNING

While PRMs offer more fine-grained supervision, directly using their scores as a reward signal for reinforcement learning can be challenging. These signals are often less reliable than final outcome verification. While this process-level reward usually can improve sample efficiency, it has not consistently resulted in better final performance compared to well-tuned policies trained with outcome-based verifiable feedback.

In contrast, a more robust application for process-level feedback is emerging in data selection. Recent studies consistently demonstrate that using detailed feedback to filter training data is a highly effective

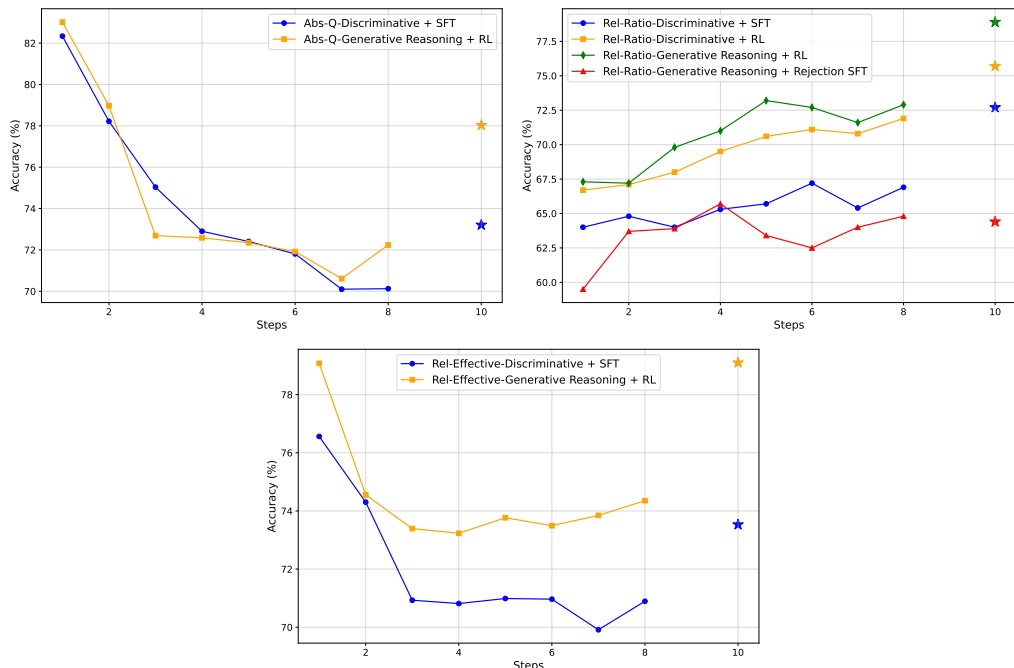

Figure 6: The test stepwise accuracy of different stepwise judges. From left to right, we plot the results of `Abs-Q`, `Rel-Ratio`, `Rel-Effective`, respectively. The stars at step 10 represent the accuracy of recognizing the final answer.

Table 8: **Data selection via Stepwise Rejection Sampling Fine-Tuning.** Our STEPWISER judge trained with RL provides better quality training data, as measured by final average test performance. The evaluation is with greedy decoding and a maximal generated length of 8192.

| Method | Learning signal | MATH500 | NM-Heldout-1K | Average ↑ |
|---|---|---|---|---|
| *Qwen2.5-7B-chunk (greedy)* | - | 75.6 | 44.6 | 60.1 |
| Outcome-based selection | - | 76.6 | 45.2 | 60.9 |
| Discriminative + SFT | `Abs-Q` | 78.4 | 45.3 | 61.8 |
| Discriminative + SFT | `Rel-Effective` | 78.2 | 45.2 | 61.7 |
| Discriminative + SFT | `Rel-Ratio` | 78.2 | 45.7 | 61.9 |
| Generative CoT + RL (STEPWISER) | `Abs-Q` | 79.0 | 46.1 | 62.5 |
| Generative CoT + RL (STEPWISER) | `Rel-Effective` | **79.4** | 46.7 | **63.0** |
| Generative CoT + RL (STEPWISER) | `Rel-Ratio` | 79.0 | **46.8** | 62.9 |

strategy, both for offline data curation (Tong et al., 2024) and online RL training (Xiong et al., 2025; Ye et al., 2025; Shrivastava et al., 2025; Xue et al., 2025). Notably, Ye et al. (2025) show that employing PRMs to select the best trajectories from a candidate pool robustly enhances final model performance.

Motivated by these findings, we evaluate our stepwise judge in a practical data selection application: Rejection Sampling Fine-tuning (RFT) (Dong et al., 2023). The goal of RFT is to improve a base policy by fine-tuning it on its own best-generated outputs. While standard RFT might select trajectories based only on final answer correctness, this coarse signal cannot differentiate between multiple valid reasoning paths that lead to the same answer.

We leverage our generative judge (STEPWISER) to provide a more fine-grained selection criterion. Specifically, from a pool of trajectories that all reach the correct final answer, we use the judge to score each individual reasoning chunk. The trajectory with the highest average chunk score is then selected as the highest-quality reasoning trace for the RFT dataset.

Table 9: Judge performance on ProcessBench, broken down by four subsets. Each subset reports Error (%), Correct (%), and F1 score (%). The final column is the average F1 across all subsets. We remark that the F1 score here is indeed the harmonic mean of the accuracies on two classes.

| Method | Learning Signal | GSM8K | | | MATH | | | Olympiad | | | Omni-MATH | | | Avg. F1 |
|---|---|---|---|---|---|---|---|---|---|---|---|---|---|---|
| | | Error | Correct | F1 | Error | Correct | F1 | Error | Correct | F1 | Error | Correct | F1 | |
| *Qwen2.5-1.5B-chunk* | | | | | | | | | | | | | | |
| Discriminative + SFT | `Abs-Q` | 26.0 | 80.0 | 39.3 | 22.2 | 57.6 | 32.1 | 14.2 | 30.2 | 19.3 | 13.2 | 28.2 | 18.0 | 27.2 |
| Discriminative + SFT | `Rel-Effect` | 28.5 | 72.0 | 40.8 | 28.6 | 53.0 | 37.2 | 16.4 | 21.8 | 18.7 | 15.8 | 27.6 | 20.1 | 29.2 |
| Discriminative + SFT | `Rel-Ratio` | 22.5 | 56.0 | 32.1 | 26.2 | 41.0 | 32.0 | 14.0 | 14.4 | 14.2 | 15.2 | 22.0 | 18.0 | 24.1 |
| Generative + CoT + RL | `Abs-Q` | 42.5 | 58.5 | 49.2 | 36.4 | 45.6 | 40.5 | 31.4 | 19.2 | 23.8 | 32.8 | 29.4 | 31.0 | 36.1 |
| Generative + CoT + RL | `Rel-Effect` | 38.5 | 64.5 | 48.2 | 37.8 | 51.6 | 43.6 | 23.2 | 21.0 | 22.1 | 24.0 | 26.8 | 25.3 | 34.8 |
| Generative + CoT + RL | `Rel-Ratio` | 35.0 | 71.0 | 46.9 | 37.8 | 50.8 | 43.4 | 27.0 | 25.6 | 26.3 | 28.0 | 28.8 | 28.4 | 36.2 |
| Gen + RL (no CoT) | `Rel-Ratio` | 28.5 | 79.5 | 42.0 | 37.0 | 51.8 | 43.2 | 24.4 | 22.8 | 23.6 | 28.6 | 28.8 | 28.7 | 34.3 |
| Gen + CoT + RL (no Chunk) | `Rel-Ratio` | 36.0 | 65.5 | 46.5 | 37.0 | 39.8 | 38.4 | 25.4 | 15.2 | 19.0 | 29.4 | 23.0 | 25.8 | 32.4 |
| *Qwen2.5-7B-chunk* | | | | | | | | | | | | | | |
| Discriminative + SFT | `Abs-Q` | 41.0 | 80.5 | 54.3 | 36.0 | 66.4 | 46.7 | 28.8 | 43.4 | 34.6 | 21.8 | 39.6 | 28.1 | 40.9 |
| Discriminative + SFT | `Rel-Effect` | 40.5 | 80.0 | 53.8 | 36.8 | 69.6 | 48.1 | 27.0 | 36.2 | 30.93 | 24.6 | 41.0 | 30.8 | 38.7 |
| Discriminative + SFT | `Rel-Ratio` | 37.5 | 78.5 | 50.8 | 36.6 | 63.8 | 46.5 | 24.0 | 35.8 | 28.7 | 24.2 | 35.6 | 28.8 | 38.7 |
| Generative + CoT + RL | `Abs-Q` | 59.5 | 64.5 | 61.9 | 63.2 | 59.0 | 61.0 | 53.0 | 44.6 | 48.4 | 44.4 | 43.4 | 43.9 | 53.8 |
| Generative + CoT + RL | `Rel-Effect` | 70.5 | 74.5 | 72.4 | 69.2 | 67.4 | 68.3 | 61.4 | 48.8 | 54.4 | 54.4 | 50.6 | 52.4 | 61.9 |
| Generative + CoT + RL | `Rel-Ratio` | 66.5 | 80.0 | 72.6 | 62.6 | 72.6 | 67.2 | 57.2 | 48.2 | 52.3 | 49.4 | 50.2 | 49.8 | 60.5 |
| *Qwen2.5-7B-chunk* Ablation | | | | | | | | | | | | | | |
| Gen + CoT + RL (no Balancing) | `Abs-Q` | 31.5 | 94.0 | 47.2 | 34.0 | 79.6 | 47.7 | 25.0 | 58.0 | 34.9 | 23.2 | 43.2 | 30.2 | 40.0 |
| Gen + CoT + RL (no Balancing) | `Rel-Effect` | 45.0 | 94.0 | 60.9 | 44.8 | 79.0 | 57.2 | 35.8 | 59.2 | 44.6 | 27.0 | 48.8 | 34.8 | 49.4 |
| Gen + CoT + RL (no Balancing) | `Rel-Ratio` | 42.5 | 95.5 | 58.8 | 41.6 | 80.2 | 54.8 | 29.8 | 65.8 | 41.0 | 29.4 | 49.6 | 36.9 | 47.9 |
| Gen + RL (no CoT) | `Rel-Ratio` | 45.5 | 82.5 | 58.7 | 37.6 | 72.0 | 49.4 | 36.0 | 47.0 | 40.8 | 40.2 | 45.6 | 42.7 | 47.9 |

Table 10: Prompt template for solver model to produce self-segmented CoTs.

---

**Prompt Template for Policy Model**

You are a helpful assistant designed to solve mathematical problems step-by-step. Your task is to think step-by-step and provide a detailed solution process following a specific format.
You **MUST** follow the following format:

1. Enclose each part of your step-by-step reasoning within `<chunk>` and `</chunk>` tags.

2. After completing the chain-of-thought reasoning, provide the final answer within `\boxed{{}}`.

Ensure strict compliance with this format for every response.

---

**Problem:** `{problem}`

---

The results of this approach, presented in Table 8, confirm the benefits of our fine-grained selection signal. Models fine-tuned on data selected by our generative judge achieve the highest average performance (63.0%). This surpasses not only the original base model (60.1%) and simple outcome-based selection (60.9%), but also data selected by the discriminative baseline judge (61.9%). This result strongly demonstrates that our judge's nuanced, step-level evaluations provide a superior signal for identifying high-quality reasoning data for model training.

# D  TEMPLATE, EXAMPLE, AND ADDITIONAL TABLES

Table 11: A representative example illustrating CoT segmentation methods based on a response from Qwen2.5-1.5B-it. Standard splitting by newlines (left) produces too many chunks without a clear logical step. For example, chunk 3 and chunk 4 are logically a whole, but they are separated, resulting in poor independence of each step and the inability to form a complete thinking unit.

| Split by \n\n | LLM-based self-segmentation (Chunks-of-Thought) |
|---|---|
| **chunk 1**
To find the value of $f(5)$ for the given function $$f(x) = \begin{cases} x - 3, & x \geq 6 \\ f(f(x+5)), & x < 6 \end{cases},$$ we need to follow the recursive definition of the function for $x < 6$. | **chunk 1 (Initial Expansion)**
To find the value of $f(5)$ for the given function $$f(x) = \begin{cases} x - 3, & x \geq 6 \\ f(f(x+5)), & x < 6 \end{cases},$$ we need to follow the recursive definition of the function for $x < 6$.
Since $5 < 6$, we use the second case of the piecewise function: $$f(5) = f(f(5+5)) = f(f(10)).$$ |
| **chunk 2**
Since $5 < 6$, we use the second case of the piecewise function: $$f(5) = f(f(5+5)) = f(f(10)).$$ | |
| **chunk 3**
Now, we need to evaluate $f(10)$. Since $10 \geq 6$, we use the first case of the piecewise function: $$f(10) = 10 - 3 = 7.$$ | **chunk 2 (Calculate Inner Function and Substitute)**
Now, we need to evaluate $f(10)$. Since $10 \geq 6$, we use the first case of the piecewise function: $$f(10) = 10 - 3 = 7.$$ So, we have: $$f(5) = f(7).$$ |
| **chunk 4**
So, we have: $$f(5) = f(7).$$ | |
| **chunk 5**
Next, we need to evaluate $f(7)$. Again, since $7 \geq 6$, we use the first case of the piecewise function: $$f(7) = 7 - 3 = 4.$$ | **chunk 3 (Final Calculation)**
Next, we need to evaluate $f(7)$. Again, since $7 \geq 6$, we use the first case of the piecewise function: $$f(7) = 7 - 3 = 4.$$ Thus, we have: $$f(5) = 4.$$ The final answer is: $$\boxed{4}$$ |
| **chunk 6**
Thus, we have: $$f(5) = 4.$$ | |
| **chunk 7**
The final answer is: $$\boxed{4}$$ . | |

Table 12: A representative example illustrating CoT segmentation methods based on a response from Qwen2.5-1.5B-it. Standard splitting by newlines (left) produces too many chunks without a clear logical step.

| Split by \n\n | LLM-based self-segmentation (Chunks-of-Thought) |
|---|---|
| **chunk 1**
To solve the given problem, we start by expressing $z$ in its standard form $z = x + yi$, where $x$ and $y$ are real numbers. The given equation is:

**chunk 2**
$$\left\|\frac{z^2+1}{z+i}\right\| + \left\|\frac{z^2+4i-3}{z-i+2}\right\| = 4.$$

**chunk 3**
First, let's express the magnitudes of the complex numbers. For the first term, we have:

**chunk 4**
$$\left\|\frac{z^2+1}{z+i}\right\| = \left\|\frac{(x+yi)^2+1}{x+yi+i}\right\|.$$

**chunk 5**
Since $(x+yi)^2 = x^2 - y^2 + 2xyi$, we get $z^2 + 1 = x^2 - y^2 + 2xyi + 1 = (x^2 - y^2 + 1) + 2xyi$. Therefore, the magnitude is:

**chunk 6**
$$\left\|(x^2 - y^2 + 1) + 2xyi + i(x+yi)\right\|$$
$$= \left\|(x^2 - y^2 + 1 + x + y) + (2xy + x + y)i\right\|.$$

**chunk 7**
This magnitude can be simplified to the square root of the sum of the squares of the real and imaginary parts:

**chunk 8**
$$\sqrt{(x^2 - y^2 + 1 + x + y)^2 + (2xy + x + y)^2}.$$

**chunk 9**
Similarly, for the second term, we have:

**chunk 10**
$$\left\|\frac{z^2+4i-3}{z-i+2}\right\| = \left\|\frac{(x+yi)^2+4i-3}{x+yi-i+2}\right\|.$$

**chunk 11**
Since $z^2 + 4i - 3 = x^2 - y^2 + 2xyi + 4i - 3 = (x^2 - y^2 - 3) + (2xy + 4)i$, we get:

**chunk 12**

$$z - i + 2 = x + (y - 1) = x + yi - i + 2 = (x + 2) + yi.$$

**chunk 13**
Using the same logic as above, the magnitude is:

...

**chunk 21**
$$\boxed{0}.$$ | **chunk 1 (Express the Equation in Terms of Real and Imaginary Parts)**
To solve the given problem, we start by expressing $z$ in its standard form $z = x + yi$, where $x$ and $y$ are real numbers. The given equation is:

$$\left\|\frac{z^2+1}{z+i}\right\| + \left\|\frac{z^2+4i-3}{z-i+2}\right\| = 4.$$

**chunk 2 (Simplify the first Term)**
First, let's express the magnitudes of the complex numbers. For the first term, we have:

$$\left\|\frac{z^2+1}{z+i}\right\| = \left\|\frac{(x+yi)^2+1}{x+yi+i}\right\|.$$

Since $(x+yi)^2 = x^2 - y^2 + 2xyi$, we get $z^2 + 1 = x^2 - y^2 + 2xyi + 1 = (x^2 - y^2 + 1) + 2xyi$. Therefore, the magnitude is:

$$\left\|(x^2 - y^2 + 1) + 2xyi + i(x+yi)\right\|$$
$$= \left\|(x^2 - y^2 + 1 + x + y) + (2xy + x + y)i\right\|.$$

This magnitude can be simplified to the square root of the sum of the squares of the real and imaginary parts:

$$\sqrt{(x^2 - y^2 + 1 + x + y)^2 + (2xy + x + y)^2}.$$

**chunk 3 (Simplify the Second Term and Formulate Equations)**
Similarly, for the second term, we have:

$$\left\|\frac{z^2+4i-3}{z-i+2}\right\| = \left\|\frac{(x+yi)^2+4i-3}{x+yi-i+2}\right\|.$$

Since $z^2 + 4i - 3 = x^2 - y^2 + 2xyi + 4i - 3 = (x^2 - y^2 - 3) + (2xy + 4)i$, we get:

$$z - i + 2 = x + (y - 1) = x + yi - i + 2 = (x + 2) + yi.$$

Using the same logic as above, the magnitude is:

$$\sqrt{(x^2 - y^2 - 3 + x + y + 2)^2 + (2xy + 4 + x + y)^2}$$
$$= \sqrt{(x^2 - y^2 - 1 + x + y)^2 + (2xy + x + y + 4)^2}.$$

**chunk 4 (Solve the System of Equations to Find z)**
Given that these two magnitudes sum to 4, and recognizing that the equality $a^2 + b^2 = 4$ indicates that each term must separately be 2 (because the maximum magnitude of any complex number is 2), we derive:

$$(x^2 - y^2 + 1 + x + y)^2 + (2xy + x + y)^2 = 4,$$
$$\left(x^2 - y^2 - 3 + x + y + 2\right)^2 + (2xy + 4 + x + y)^2 = 4.$$

This leads to $x^2 - y^2 + x = 1$ and $x^2 - y^2 - x = -1$. Solving these equations simultaneously, we get $x = 1$ and $y = 0$, so $z = 1$.

**chunk 5 (Calculate the Final Minimum Value)**
Finally, we need to find the minimum value of $\|z - 1\|$. This simplifies to:

$$\|z - 1\| = \|1 - 1\| = 0.$$

Thus, the minimum value of $\|z - 1\|$ is:

$$\boxed{0}.$$ |

Table 13: A representative example illustrating CoT segmentation methods based on a response from Qwen2.5-1.5B-it. Standard splitting by newlines (left) produces too many chunks without a clear logical step.

| Split by \n\n | LLM-based self-segmentation (Chunks-of-Thought) |
|---|---|
| **chunk 1**
Let's start by analyzing the given conditions: 1. When 5655 is divided by $N$, the remainder is 11. This can be written as:

$$5655 = qN + 11$$

where $q$ is some integer. | |
| **chunk 2**
2. When 5879 is divided by $N$, the remainder is 14. This can be written as:

$$5879 = mN + 14$$

where $m$ is some integer. | **chunk 1 (Formulating the Equations)**
Let's start by analyzing the given conditions: 1. When 5655 is divided by $N$, the remainder is 11. This can be written as:

$$5655 = qN + 11$$

where $q$ is some integer.
2. When 5879 is divided by $N$, the remainder is 14. This can be written as:

$$5879 = mN + 14$$

where $m$ is some integer. |
| **chunk 3**
We can express these equations as:

$$5655 - 11 = 5644 = Nq$$

$$5879 - 14 = 5865 = Nm$$ | We can express these equations as:

$$5655 - 11 = 5644 = Nq$$

$$5879 - 14 = 5865 = Nm$$ |
| **chunk 4**
We know that $N$ is a two-digit positive integer. Therefore, we need to find a value of $N$ that satisfies the above conditions for integer $q$ and $m$. | **chunk 2 (Finding Common Divisors via Factorization)**
We know that $N$ is a two-digit positive integer. Therefore, we need to find a value of $N$ that satisfies the above conditions for integer $q$ and $m$. |
| **chunk 5**
First, let's determine the possible values of $N$ by factoring 5644 and 5865:

$$5644 = 2^2 \cdot 7 \cdot 11^2$$

$$5865 = 3 \cdot 5 \cdot 13^2$$ | First, let's determine the possible values of $N$ by factoring 5644 and 5865:

$$5644 = 2^2 \cdot 7 \cdot 11^2$$

$$5865 = 3 \cdot 5 \cdot 13^2$$ |
| **chunk 6**
We need to find a common divisor for both 5644 and 5865. Let's check the common divisors:

Common divisors are: $1, 7, 11, 143$ | We need to find a common divisor for both 5644 and 5865. Let's check the common divisors:

Common divisors are: $1, 7, 11, 143$ |
| **chunk 7**
Since $N$ is a two-digit positive integer, we consider the possible values:

$$N = 11 \quad \text{(if 11 is a divisor of both 5644 and 5865)}$$ | **chunk 3 (Verifying the Candidate N)**
Since $N$ is a two-digit positive integer, we consider the possible values:

$$N = 11 \quad \text{(if 11 is a divisor of both 5644 and 5865)}$$

Let's test $N = 11$: 1. Check if 11 divides 5644 exactly: |
| **chunk 8**
Let's test $N = 11$: 1. Check if 11 divides 5644 exactly:

$$5644 = 11 \times 513 \quad \text{Yes}$$

2. Check if 11 divides 5865 exactly:

$$5865 = 11 \times 533 \quad \text{Yes}$$ | $$5644 = 11 \times 513 \quad \text{Yes}$$

2. Check if 11 divides 5865 exactly:

$$5865 = 11 \times 533 \quad \text{Yes}$$

Since $N = 11$ divides 5644 and 5865 exactly, $N = 11$ satisfies the conditions. |
| **chunk 9**
Since $N = 11$ divides 5644 and 5865 exactly, $N = 11$ satisfies the conditions. Now we have:

$$Nq = 5644 \quad \text{and} \quad Nm = 5865$$ | **chunk 4 (Final Calculation)**
Now we have:
$$Nq = 5644 \quad \text{and} \quad Nm = 5865$$

Using the Law of Cosines (not needed for this case, but for completeness): |
| **chunk 10**
Using the Law of Cosines (not needed for this case, but for completeness):

$$\cos(\theta) = \frac{a^2 + b^2 - c^2}{2ab}$$

where $a = 5644$, $b = 5865$, and $c$ is the hypotenuse (since 11 * 11 * 1 is our $N$). | $$\cos(\theta) = \frac{a^2 + b^2 - c^2}{2ab}$$

where $a = 5644$, $b = 5865$, and $c$ is the hypotenuse (since 11 * 11 * 1 is our $N$).
Sum of the digits of $N$:
$$1 + 1 = 2$$
Thus, the sum of the digits of $N$ is:

$$\boxed{2}$$ |
| **chunk 11**
Sum of the digits of $N$:
$$1 + 1 = 2$$ | |
| **chunk 12**
Thus, the sum of the digits of $N$ is:

$$\boxed{2}$$ | |

Table 14: An illustrative example from the Process-Bench dataset, where the ground-truth label is "correct." The judge model is STEPWISER with `Rel-Ratio` and Qwen2.5-7B-chunk.

---

**Example**

**Mathematical Problem**
Cindy's math and science books weigh 2 pounds each. Her French book weighs 4 pounds and her English book weighs 3 pounds. Her history book weighs twice as much as her English book. If Cindy carries all of her books at once, what will be the total weight of the books she is carrying?

---

**Historical Reasoning Path**

- To determine the total weight of all Cindy's books, we need to calculate the weight of each book individually and then sum these weights.

- **Math and Science Books:**
  - Each math book weighs 2 pounds.
  - Each science book weighs 2 pounds.
  - Cindy has 2 math books and 2 science books.
  - Total weight of math books: $2 \times 2 = 4$ pounds.
  - Total weight of science books: $2 \times 2 = 4$ pounds.
  - Combined weight: $4 + 4 = 8$ pounds.

- **French Book:**
  - The French book weighs 4 pounds.

---

**New Reasoning Chunk**
Third, for the English book:

- The English book weighs 3 pounds.

---

**Judgment**
To determine if the "New Reasoning_chunk" contains any explicit errors, we need to analyze the information given in the problem and the reasoning presented.
The new reasoning chunk states: Third, for the English book:

- The English book weighs 3 pounds.

From the problem statement, we know:

- The French book weighs 4 pounds.

- The English book weighs 3 pounds.

- The history book weighs twice as much as the English book, so the history book weighs $2 \times 3 = 6$ pounds.

- The math and science books each weigh 2 pounds, and Cindy has 2 math books and 2 science books.

The new reasoning chunk simply states that the English book weighs 3 pounds, which is directly given in the problem statement. Therefore, this statement is correct and logical. Thus, the final judgment is:

$$\boxed{Positive}$$

.

---

Table 15: An illustrative example from the Process-Bench dataset, where the ground-truth label is "correct." The judge model is STEPWISER with `Rel-Ratio` and Qwen2.5-7B-chunk.

---

**Example**

**Mathematical Problem**
An $n$-label is a permutation of the numbers 1 through $n$. For example, $J = 35214$ is a 5-label and $K = 132$ is a 3-label. For a fixed positive integer $p$, where $p \leq n$, consider consecutive blocks of $p$ numbers in an $n$-label. For example, when $p = 3$ and $L = 263415$, the blocks are 263, 634, 341, and 415. We can associate to each of these blocks a $p$-label that corresponds to the relative order of the numbers in that block. For $L = 263415$, we get the following:

$$\underline{263}\,415 \rightarrow 132; \quad 2\underline{634}15 \rightarrow 312; \quad 26\underline{341}5 \rightarrow 231; \quad 263\underline{415} \rightarrow 213$$

Moving from left to right in the $n$-label, there are $n - p + 1$ such blocks, which means we obtain an $(n - p + 1)$-tuple of $p$-labels. For $L = 263415$, we get the 4-tuple $(132, 312, 231, 213)$. We will call this $(n - p + 1)$-tuple the $p$-signature of $L$ (or signature, if $p$ is clear from the context) and denote it by $S_p[L]$; the $p$-labels in the signature are called windows. For $L = 263415$, the windows are 132, 312, 231, and 213, and we write
$$S_3[263415] = (132, 312, 231, 213)$$
More generally, we will call any $(n - p + 1)$-tuple of $p$-labels a $p$-signature, even if we do not know of an $n$-label to which it corresponds (and even if no such label exists). A signature that occurs for exactly one $n$-label is called *unique*, and a signature that doesn't occur for any $n$-labels is called *impossible*. A *possible* signature is one that occurs for at least one $n$-label.
In this power question, you will be asked to analyze some of the properties of labels and signatures.
**Compute the 3-signature for 52341.**

---

**Historical Reasoning Path**
To compute the 3-signature of the 5-label $L = 52341$, we need to identify all the consecutive blocks of 3 numbers in the label and then convert each block into its corresponding 3-label based on the relative order of the numbers.

First, identify the first block: 523. The relative order of the numbers 5, 2, and 3 is 3, 1, and 2 respectively. Therefore, the 3-label for this block is 312.

Second, identify the second block: 234. The relative order of the numbers 2, 3, and 4 is 1, 2, and 3 respectively. Therefore, the 3-label for this block is 123.

---

**New Reasoning Chunk**
Third, identify the third block: 341. The relative order of the numbers 3, 4, and 1 is 2, 3, and 1 respectively. Therefore, the 3-label for this block is 231.

---

**Judgment**
To determine if the "New Reasoning_chunk" contains any explicit errors, we need to verify the logic for identifying the 3-label of the block 341 in the 5-label 52341.
The new reasoning chunk states:

> Third, identify the third block: 341. The relative order of the numbers 3, 4, and 1 is 2, 3, and 1 respectively. Therefore, the 3-label for this block is 231.

Let's check the relative order of the numbers in the block 341:

- The number 3 is greater than 1 but less than 4.
- The number 4 is greater than both 3 and 1.
- The number 1 is the smallest.

Thus, the relative order of the numbers 3, 4, and 1 is indeed 2, 3, and 1 respectively. This means the 3-label for the block 341 is 231.

Since the reasoning follows correctly from the definition of a 3-label, the "New Reasoning_chunk" does not contain any explicit errors.

Therefore, the final judgment is:

Positive

---

Table 16: An illustrative example from the Process-Bench dataset, where the ground-truth label is "incorrect." The judge model is STEPWISER with `Rel-Ratio` and Qwen2.5-7B-chunk. We highlight the key parts of the reasoning trajectory in red.

---

**Example**

**Mathematical Problem**
There are 60 empty boxes $B_1, \ldots, B_{60}$ in a row on a table and an unlimited supply of pebbles. Given a positive integer $n$, Alice and Bob play the following game.
In the first round, Alice takes $n$ pebbles and distributes them into the 60 boxes as she wishes. Each subsequent round consists of two steps:
(a) Bob chooses an integer $k$ with $1 \le k \le 59$ and splits the boxes into the two groups $B_1, \ldots, B_k$ and $B_{k+1}, \ldots, B_{60}$.
(b) Alice picks one of these two groups, adds one pebble to each box in that group, and removes one pebble from each box in the other group.
Bob wins if, at the end of any round, some box contains no pebbles. Find the smallest $n$ such that Alice can prevent Bob from winning.

---

**Historical Reasoning Path**
To find the smallest $n$ such that Alice can prevent Bob from winning, we need to understand the constraints imposed by Bob's moves and how Alice can counter them. Let's consider the minimum number of pebbles each box must have after the first round. Since there are 60 boxes and Alice distributes $n$ pebbles among them, the minimum number of pebbles each box will get is $\lfloor \frac{n}{60} \rfloor$, where $\lfloor x \rfloor$ represents the greatest integer less than or equal to $x$.

---

**New Reasoning Chunk**
Now, let's analyze Bob's move in each round. In step (a), he splits the boxes into two groups of size $k$ and $60 - k$. In step (b), Alice adds a pebble to each box in the chosen group and removes a pebble from each box in the other group. For Alice to prevent Bob from winning, each box must have at least one pebble after her move. This means that the difference between the total number of pebbles in the chosen group and the other group should be at most 1.

---

**Judgment**
Let's break this down step by step:

1. **Understanding Bob's Move**: In step (a), Bob splits the boxes into two groups of size $k$ and $60 - k$. In step (b), Alice adds one pebble to each box in the chosen group and removes one pebble from each box in the other group. This means that the total number of pebbles in the chosen group increases by $k$ and the total number of pebbles in the other group decreases by $60 - k$.

2. **Condition for Alice to Prevent Bob from Winning**: For Alice to prevent Bob from winning, each box must have at least one pebble after her move. However, the statement "the difference between the total number of pebbles in the chosen group and the other group should be at most 1" is not necessarily true. The correct condition would be that the difference in the total number of pebbles between the two groups after Alice's move should not cause any box to have zero pebbles.

The reasoning in the new reasoning chunk assumes that the difference in the number of pebbles between the two groups should be at most 1, which is not necessarily the case. The correct condition would be that after Alice's move, the number of pebbles in each box should be at least 1, not necessarily that the difference in the total number of pebbles between the two groups should be at most 1.

Therefore, the "New Reasoning_chunk" contains explicit errors. The correct interpretation of the condition for Alice to prevent Bob from winning is not correctly stated in the new reasoning chunk.

The final judgment is:

$$\boxed{\text{Negative}}$$

---

Table 17: An illustrative example from the Process-Bench dataset, where the ground-truth label is "incorrect." The judge model is STEPWISER with `Rel-Ratio` and Qwen2.5-7B-chunk. We highlight the key parts of the reasoning trajectory in red.

---

**Example**

**Mathematical Problem**

Let $n$ and $k$ be fixed positive integers of the same parity, $k \geq n$. We are given $2n$ lamps numbered 1 through $2n$; each of them can be on or off. At the beginning all lamps are off. We consider sequences of $k$ steps. At each step one of the lamps is switched (from off to on or from on to off).

Let $N$ be the number of $k$-step sequences ending in the state: lamps $1, \ldots, n$ on, lamps $n+1, \ldots, 2n$ off.

Let $M$ be the number of $k$-step sequences leading to the same state and not touching lamps $n + 1, \ldots, 2n$ at all.

Find the ratio $N/M$.

---

**Historical Reasoning Path**

To solve the problem, we need to determine the number of $k$-step sequences that end in the state where lamps $1, \ldots, n$ are on and lamps $n + 1, \ldots, 2n$ are off, and compare it to the number of such sequences that do not touch lamps $n + 1, \ldots, 2n$ at all.

---

**New Reasoning Chunk**

First, we determine $N$. The total number of lamps is $2n$, and we need exactly $n$ of them to be on and $n$ to be off. Since the sequence ends with lamps $1, \ldots, n$ on and $n+1, \ldots, 2n$ off, we need to switch exactly $n$ times among the first $n$ lamps and the remaining $k - n$ times among the last $n$ lamps. However, since the last $n$ lamps are always off, these $k - n$ switches do not affect their state. Therefore, we only need to count the number of ways to choose $n$ steps out of $k$ to switch the first $n$ lamps. This is given by the binomial coefficient:

$$N = \binom{k}{n}.$$

---

**Judgment**

To determine if the "New Reasoning_chunk" contains any explicit errors, we need to carefully analyze the statement and check its logic step by step.

The reasoning chunk suggests that $N$, the number of $k$-step sequences ending in the state where lamps $1, \ldots, n$ are on and lamps $n + 1, \ldots, 2n$ are off, is given by the binomial coefficient $\binom{k}{n}$. However, this conclusion is not necessarily correct because it does not take into account the specific sequence of switching the lamps.

The correct approach to find $N$ is to consider that we need exactly $n$ switches among the first $n$ lamps to turn them on, and the remaining $k - n$ switches can be any combination of the $n$ lamps (including possibly switching some lamps on and off multiple times, but ensuring that at the end, each of the first $n$ lamps is on). The number of such sequences is not simply $\binom{k}{n}$, because it does not account for the specific order in which the switches occur.

In fact, the correct way to calculate $N$ is more complex and involves considering the parity of $k$ and $n$ (which are the same) and the specific sequences of switching. The reasoning chunk simplifies the problem incorrectly by assuming that any $n$ switches among the first $n$ lamps will result in all of them being on, without considering the constraints of the problem.

Therefore, the "New Reasoning_chunk" contains explicit errors. The correct reasoning would involve a more detailed analysis of the sequences of switches, taking into account the parity and the specific requirements of the problem.

The final judgment is:

| Negative |
|---|

---

Table 18: An illustrative example from the Process-Bench dataset, where the ground-truth label is "incorrect." The judge model is STEPWISER with `Rel-Ratio` and Qwen2.5-7B-chunk. We highlight the key parts of the reasoning trajectory in red.

---

**Example**

**Mathematical Problem**
Determine all functions $f$ from the set of positive integers into the set of positive integers such that for all $x$ and $y$ there exists a non degenerated triangle with sides of lengths

$$x, \quad f(y) \text{ and } f(y + f(x) - 1).$$

---

**Historical Reasoning Path**
To determine all functions $f$ from the set of positive integers into the set of positive integers such that for all $x$ and $y$ there exists a non-degenerate triangle with sides of lengths $x$, $f(y)$, and $f(y + f(x) - 1)$, we need to use the triangle inequality. The triangle inequality states that the sum of the lengths of any two sides of a triangle must be greater than the length of the remaining side. Therefore, we have the following three inequalities: First, $x + f(y) > f(y + f(x) - 1)$ Second, $x + f(y + f(x) - 1) > f(y)$ Third, $f(y) + f(y + f(x) - 1) > x$
We will analyze these inequalities one by one.
### Step 1: Analyze the inequality $x + f(y) > f(y + f(x) - 1)$
This inequality can be rewritten as:

$$f(y + f(x) - 1) < x + f(y)$$

This tells us that $f(y + f(x) - 1)$ is strictly less than $x + f(y)$.

---

**New Reasoning Chunk**
### Step 2: Analyze the inequality $x + f(y + f(x) - 1) > f(y)$
This inequality can be rewritten as:

$$f(y + f(x) - 1) > f(y) - x$$

This tells us that $f(y + f(x) - 1)$ is strictly greater than $f(y) - x$. Since $f(y)$ and $x$ are positive integers, $f(y) - x$ could be negative or zero, but $f(y + f(x) - 1)$ must be a positive integer. Therefore, the smallest possible value for $f(y) - x$ is 0, which means $f(y) \geq x$.

---

**Judgment**
To determine if the "New Reasoning_chunk" contains any explicit errors, we need to analyze the given inequality $x + f(y + f(x) - 1) > f(y)$ and see if it leads to the conclusion that $f(y) \geq x$.

Starting with the inequality:
$$x + f(y + f(x) - 1) > f(y)$$
we can rearrange it to:
$$f(y + f(x) - 1) > f(y) - x$$
Since $f(y)$ and $x$ are positive integers, $f(y) - x$ could be negative or zero. However, $f(y + f(x) - 1)$ must be a positive integer. This means that $f(y) - x$ must be less than $f(y + f(x) - 1)$, but it does not necessarily mean that $f(y) \geq x$. The inequality $f(y + f(x) - 1) > f(y) - x$ only tells us that $f(y) - x$ is less than a positive integer, which does not imply that $f(y)$ itself must be at least $x$.

Therefore, the reasoning in the new reasoning chunk contains an explicit error in its conclusion that $f(y) \geq x$. The correct interpretation of the inequality $f(y + f(x) - 1) > f(y) - x$ does not necessarily lead to $f(y) \geq x$.

The final judgment is:

$$\boxed{\text{Negative}}$$

---

Table 19: An illustrative example from the Process-Bench dataset, where the ground-truth label is "incorrect." The judge model is STEPWISER with `Rel-Ratio` and Qwen2.5-7B-chunk. We highlight the key parts of the reasoning trajectory in red.

---

**Example**

**Mathematical Problem**
How many ways are there to put 4 balls in 3 boxes if the balls are distinguishable but the boxes are not?

---

**Historical Reasoning Path**
To solve this problem, we need to consider the number of ways to distribute 4 distinguishable balls into 3 indistinguishable boxes.

Step 1: Counting the total number of distributions if the boxes were distinguishable. If the boxes were distinguishable, each ball would have 3 choices (3 boxes). Since there are 4 distinguishable balls, the total number of distributions would be $3^4 = 81$. However, since the boxes are indistinguishable, we need to adjust for overcounting.

Step 2: Identifying the cases where the boxes are indistinguishable. There are three main cases where the boxes are indistinguishable:

First, all 4 balls in one box (1 way).
Second, 3 balls in one box and 1 ball in another box (4 ways, since the balls are distinguishable and we can choose any of the 4 balls to be alone).
Third, 2 balls in one box and 2 balls in another box (6 ways, since there are 6 pairs of balls that can be in the same box). However, these cases also include some overcounting since the boxes are indistinguishable. We need to adjust for this overcounting.

---

**New Reasoning Chunk**
Step 3: Adjusting for overcounting. In case 2, when we have 3 balls in one box and 1 ball in another box, we counted each distribution 3 times (since there are 3 boxes and we could swap the lone ball with the other balls). So, we divide by 3 for this case. In case 3, when we have 2 balls in one box and 2 balls in another box, we counted each distribution $6/2 = 3$ times (since there are 3 pairs of boxes). So, we divide by 3 for this case. Now we have:

- Case 1: 1 way
- Case 2: 4 / 3 ways
- Case 3: 6 / 3 ways

---

**Judgment**
To determine if the "New Reasoning_chunk" contains any explicit errors, we need to carefully analyze the reasoning step by step. The new reasoning chunk states: In case 2, when we have 3 balls in one box and 1 ball in another box, we counted each distribution 3 times (since there are 3 boxes and we could swap the lone ball with the other balls). So, we divide by 3 for this case. In case 3, when we have 2 balls in one box and 2 balls in another box, we counted each distribution $6/2 = 3$ times (since there are 3 pairs of boxes). So, we divide by 3 for this case...

Let's evaluate each case:

**Case 2: 3 balls in one box and 1 ball in another box**
When we have 3 balls in one box and 1 ball in another box, we do indeed count each distribution 3 times because we can place the single ball in any one of the 3 boxes. Therefore, the correct number of distinct distributions for this case is 4 (the number of ways to choose which ball will be the single ball) divided by 3, but this division is incorrect because it assumes that each distribution is counted exactly 3 times, which is not necessarily true. The correct approach would be to consider the different ways to assign the balls without overcounting due to the indistinguishability of the boxes. The correct count for this case is actually 4, not $4/3$.
**Case 3: 2 balls in one box and 2 balls in another box**
When we have 2 balls in one box and 2 balls in another box, we need to count the number of ways to partition the 4 balls into two groups of 2. The number of ways to choose 2 balls out of 4 is $\binom{4}{2} = 6$, but since the boxes are indistinguishable, we have counted each distribution twice (once for each way to assign the groups to the boxes). Therefore, the correct number of distinct distributions for this case is $6/2 = 3$, which the reasoning chunk correctly states.

Given the above analysis, the reasoning in the new reasoning chunk contains an explicit error in its calculation for Case 2. Therefore, the final judgment is:

> Negative

---

Table 20: An illustrative example of chunk-reset reasoning. The judge model is STEPWISER with `Rel-Effective` and Qwen2.5-7B-chunk. In the first attempt of chunk 3, it contains a significant algebraic mistake in the formula relating $\lambda$ and $r$.

---

**Example**

**Mathematical Problem**

8 singers participate in the arts festival, preparing to arrange $m$ performances, each time 4 of them will perform on stage, requiring that any two of the 8 singers perform together the same number of times. Please design a scheme so that the number of performances $m$ is the least.

---

**Chunk 1**

To solve this problem, we need to ensure that each pair of singers performs together in the same number of performances. This is a specific type of combinatorial design problem, and the solution can be approached using the concept of a balanced incomplete block design (BIBD).

Here are the steps to find the solution: **1. Identify the parameters of the BIBD:**

- Let ( v = 8 ) (the number of singers).
- Let ( k = 4 ) (the number of singers performing in each performance).
- Let ( r ) be the number of performances each singer participates in.
- Let ( $\lambda$ ) be the number of performances each pair of singers participates in together.
- Let ( b ) be the number of performances (the number we need to determine).

---

**Chunk 2**

**2. Use the BIBD parameters to derive equations:** The equation for the number of performances each singer participates in is ( r(k-1) = (v-1)$\lambda$ ). The equation for the number of performances is ( $b = \frac{vr}{k}$ ).

---

**Chunk 3** (first attempt): This step uses the incorrect formula $\lambda = \frac{4(r-1)}{7}$ instead of the correct one $\lambda = \frac{3r}{7}$.

**3. Determine the value of ($\lambda$):**

- From the first equation, we can express ($\lambda$) as ($\lambda = \frac{r(k-1)}{v-1}$).
- Substituting ( v = 8 ) and ( k = 4 ), we get ($\lambda = \frac{4r-4}{7} = \frac{4(r-1)}{7}$).
- Since ($\lambda$) must be an integer, ( 4(r-1) ) must be divisible by 7. The smallest positive integer ( r ) that satisfies this condition is ( r = 2 ) (since ( 4(2-1) = 4 ), and 4 is not divisible by 7, the next is ( r = 8 ) which gives ( 4(8-1) = 28 ), which is divisible by 7).
- Therefore, ($\lambda = \frac{28}{7} = 4$).

---

**Chunk 3** (second attempt)

**3.Calculate the parameters:**

- First, we need to find ($\lambda$). We know that ($r = \frac{(v-1)\lambda}{k-1}$). For ($\lambda$) to be an integer, ($v-1)\lambda$ must be divisible by ($k-1 = 3$). So, $\lambda$ must be a multiple of 3.
- Let's try ($\lambda = 1$). Then ($r = \frac{(8-1)\cdot 1}{4-1} = \frac{7}{3}$), which is not an integer.
- Let's try ($\lambda = 3$). Then ($r = \frac{(8-1)\cdot 3}{4-1} = \frac{21}{3} = 7$).
- Now, we can calculate ( b ): ($b = \frac{8\cdot 7}{4} = 14$).

---

**Chunk 4**

**4. Verify the design:**

- We need to check if we can indeed arrange 14 performances with 8 singers, 4 at a time, such that each pair of singers performs together exactly 3 times.
- his can be verified using known designs or by constructing a specific design, but for the sake of this problem, we assume the existence of such a design.

Therefore, the least number of performances m is

$$\boxed{14}.$$

---

Table 21: An illustrative example of chunk-reset reasoning. The judge model is STEPWISER with `Rel-Effective` and Qwen2.5-7B-chunk. In the first attempt of chunk 3, the model fails to verify the condition.

---

**Example**

**Mathematical Problem**
Which is the two-digit number that, when divided by the digit in the units place, gives a quotient of 9 and a remainder of 6?

---

**Chunk 1**
To solve the problem, let's break it down step by step.
**1. Identify the structure of the number and the division operation:**

- Let the two-digit number be represented as 10a + b, where a is the tens digit and b is the units digit.

- According to the problem, when this number is divided by the digit in the units place (which is b), the quotient is 9 and the remainder is 6. This can be expressed as:
$$10a + b = 9b + 6$$

---

**Chunk 2**
**2. **Simplify the equation:**

- Rearrange the equation to isolate terms involving a:
$$10a + b - 9b = 6$$

- Simplify: $10a - 8b = 6$

- Further simplify by dividing the entire equation by 2: $5a - 4b = 3$

---

**Chunk 3** (first attempt) This step fails verify the condition and stop with $b = 3$.
**3.Solve for a and b:**

- We need to find integer values of a and b that satisfy the equation 5a - 4b = 3. Since a and b are digits (0-9) and a cannot be 0 (as 10a would not be a two-digit number), let's test possible values for b.

- If b = 2:
$$5a - 4(2) = 3 \implies 5a - 8 = 3 \implies 5a = 11$$
5a = 11 does not give an integer value for a.

- If b = 1:
$$5a - 4(1) = 3 \implies 5a - 4 = 3 \implies 5a = 7$$
5a = 7 does not give an integer value for a.

- If b = 3:
$$5a - 4(3) = 3 \implies 5a - 12 = 3 \implies 5a = 15 \implies a = 3$$
This gives an integer value for a.

---

**Chunk 3** (second attempt)
**Determine possible values for a and b:**

- Since a and b are digits (0-9), we need to find values that satisfy the equation 5a - 4b = 3.

- We will test possible values for b (since b is the units digit and must be a digit from 1 to 9, as 0 would make the remainder 6 impossible).

  - Testing b = 1:
$$5a - 4(1) = 3 \implies 5a - 4 = 3 \implies 5a = 7 \implies a = \frac{7}{5} \quad \text{(not an integer)}$$

  - Testing b = 3:
$$5a - 4(3) = 3 \implies 5a - 12 = 3 \implies 5a = 15 \implies a = 3$$
a = 3 is an integer, and we need to check if b = 3 is feasible. If a = 3 and b = 3, the number is 33. We need to verify:
$$33 \div 3 = 11 \quad (\text{quotient} = 11, \text{remainder} = 0) \quad \text{(not a solution)}$$

  - Testing b = 4: $5a - 4(4) = 3 \implies 5a - 16 = 3 \implies 5a = 19 \implies a = \frac{19}{5} \quad \text{(not an integer)}$
  - Testing b = 5: $5a - 4(5) = 3 \implies 5a - 20 = 3 \implies 5a = 23 \implies a = \frac{23}{5} \quad \text{(not an integer)}$
  - Testing b = 6: $5a - 4(6) = 3 \implies 5a - 24 = 3 \implies 5a = 27 \implies a = \frac{27}{5} \quad \text{(not an integer)}$
  - Testing b = 7: $5a - 4(7) = 3 \implies 5a - 28 = 3 \implies 5a = 31 \implies a = \frac{31}{5} \quad \text{(not an integer)}$
  - Testing b = 8: $5a - 4(8) = 3 \implies 5a - 32 = 3 \implies 5a = 35 \implies a = 7$. a = 7 is an integer. So, the number is 78. We need to verify:
$$78 \div 8 = 9 \quad (\text{quotient} = 9, \text{remainder} = 6)$$
This satisfies the condition.

---

**Chunk 4**
**4. Conclusion:** The two-digit number that meets the condition is 78. Therefore, the final answer is:

$$\boxed{78}$$

Table 22: An illustrative example of chunk-reset reasoning. The judge model is STEPWISER with `Rel-Effective` and Qwen2.5-7B-chunk. The idea is to use the commutative law of multiplication and division to perform large-scale reduction, while the rejected chunks focus on the approximation for simplifying the expression.

---

**Example**

**Mathematical Problem**
Calculate: $241 \times 345 \div 678 \div 345 \times (678 \div 241)$

---

**Chunk 1 (first attempt)** we don't need the approximation for simplification
Let's break down the given expression step by step:

$$241 \times 345 \div 678 \div 345 \times \left( \frac{678}{241} \right).$$

First, let's simplify the expression inside the parentheses:

$$\left( \frac{678}{241} \right) = 2.812 \approx 2 \text{ (for simplicity, we can approximate it to 2 if dealing with integers)}$$

However, we'll use the exact value to be precise:

$$\left( \frac{678}{241} \right) = 2.812$$

---

**Chunk 1 (second attempt)**
Let's break down the given expression step by step:

$$241 \times 345 \div 678 \div 345 \times \left( \frac{678}{241} \right).$$

First, let's simplify the expression by following the operations order:
1. Calculate the division inside the parentheses first:

$$\frac{678}{241} = 2.8 \quad \text{(approximately, but we don't need the approximation for simplification here, just the fraction)}$$

So the expression becomes:

$$241 \times 345 \div 678 \div 345 \times \frac{678}{241}$$

---

**Chunk 2** (first attempt)
2. Simplify the expression by canceling out common factors:

- The 345 in the numerator and one of the denominators cancel out:

$$241 \times 345 \div 678 \div 345 \times \frac{678}{241} = 241 \times \frac{678}{678} \times \frac{678}{241}$$

- The 678 in the numerator and one of the denominators cancel out:

$$241 \times \frac{678}{678} \times \frac{678}{241} = 241 \times 1 \times \frac{678}{241}$$

- The 241 in the numerator and the denominator cancel out:

$$241 \times 1 \times \frac{678}{241} = 678 \times 1 = 678$$

---

**Chunk 2** (second attempt)
2. Notice that $345 \div 345 = 1$, so we can simplify the expression to:

$$241 \times 1 \div 678 \times \frac{678}{241}$$

$$= 241 \div 678 \times \frac{678}{241}$$

---

**Chunk 3**
3. Now, we can cancel out 678 in the numerator and the denominator:

$$241 \times \left( \frac{1}{678} \times \frac{678}{241} \right)$$

$$= 241 \times \left( \frac{1}{241} \right)$$

$$= 241 \times \frac{1}{241} = 1$$

---

**Chunk 4**
Therefore, the final answer is:

$$\boxed{1}$$

---

