# OpenReview forum: "StepWiser: Stepwise Generative Judges for Wiser Reasoning"
_ICLR.cc/2026/Conference — ICLR 2026 Conference Withdrawn Submission_

### Official Review · Reviewer_vrW4 · 2025-10-28

**Soundness:** 3
**Presentation:** 3
**Contribution:** 3
**Rating:** 6
**Confidence:** 4

**Summary:**

This paper introduces STEPWISER, a novel generative judge designed to supervise the multi-step reasoning of LLMs. Unlike traditional Process Reward Models (PRMs) that act as simple classifiers, STEPWISER reframes evaluation as a reasoning task itself. It first generates its own "meta-reasoning" (a Chain-of-Thought explaining its analysis) before delivering a final verdict on a reasoning step.Key contributions include:
1. A "Chunks-of-Thought" self-segmentation technique to create logically coherent reasoning steps.
2. An RL training framework (using GRPO) where the judge is rewarded based on the relative outcome of Monte-Carlo rollouts (i.e., whether a step improves the chance of success, $\Delta Q^{\pi}$).
3. Demonstrating that this generative, RL-trained judge significantly outperforms SFT baselines on ProcessBench, enables effective inference-time self-correction ("Chunk-Reset Reasoning"), and improves training data selection.

**Strengths:**

1. The paper creatively shifts the paradigm from a simple classification task (discriminative PRMs) to a generative reasoning task, forcing the judge to "meta-reason" by producing a CoT.
2. The authors provide comprehensive ablations that clearly isolate the individual performance gains from using RL over SFT, generative CoT over discriminative formats, and dataset balancing. The practical utility is convincingly shown through superior results in both inference-time search ("Chunk-Reset Reasoning") and RFT data selection, proving the judge's effectiveness.
3. This paper is well-written and structured. Figure 1 provides a clear, high-level overview of the entire pipeline, and concepts like "Chunks-of-Thought" are well-defined and motivated.

**Weaknesses:**

1. The paper's primary weakness is its incremental novelty. The core components (generative verifiers, MC rollouts, RL training) have been individually explored in recent works. However, the authors provide a comprehensive and systematic integration of this entire pipeline. I will give a positive view of its contribution.
2. Significant Computational and Latency Overheads. The multi-stage pipeline—requiring extensive MC rollouts for data annotation (14 days) followed by a full online RL run (5 days)—makes replication and practical application extremely expensive.
3. Inference Latency: Chunk-Reset Reasoning requires running the judge model after every generated chunk and potentially re-sampling up to 5 times. This introduces substantial latency, making it unsuitable for real-time applications. The high "Rejected length" in Table 4  confirms that this re-sampling occurs frequently. The authors should discuss these practical limitations and potential trade-offs (e.g., performance vs. compute).

**Questions:**

See weakness.

---

### Official Review · Reviewer_wzqt · 2025-10-30

**Soundness:** 2
**Presentation:** 2
**Contribution:** 2
**Rating:** 4
**Confidence:** 3

**Summary:**

The paper addresses the supervision of intermediate steps in multi-step reasoning. Traditional Process Reward Models (PRMs) are typically designed as discriminative classifiers, which assign scores to each reasoning step but fail to explain the rationale behind their judgments. Moreover, their dependence on static supervised fine-tuning (SFT) datasets limits both generalization and interpretability.

To overcome these limitations, the authors propose reframing the judgment of reasoning itself as a reasoning task—allowing the model to “reason about reasoning”, i.e., to generate its own chain of thought before delivering a final verdict.

Core Contribution: The STEPWISER Framework
STEPWISER is a generative judge designed for evaluating intermediate reasoning steps. It is trained via reinforcement learning (RL) to produce well-grounded, explanatory judgments after analyzing reasoning chunks. The framework consists of three main components:

1. Self-Segmentation of Reasoning Chains – enabling the model to divide its reasoning process into coherent “chunks of thought.”

2. Monte Carlo–based Stepwise Labeling – using relative outcome rollouts to assign dynamic rewards to reasoning chunks.

3. RL Training of a Generative Judge – teaching the model to reason about the quality of each chunk through its own chain-of-thought before producing a verdict.

**Strengths:**

This paper presents a well-motivated and technically sound contribution to the emerging area of process-level reward modeling for reasoning-intensive LLMs. Its central innovation lies in reframing stepwise evaluation as a generative reasoning task, where the judge model is trained to “reason about reasoning.” The proposed STEPWISER framework is conceptually elegant and empirically convincing: it combines (1) self-segmentation of chains-of-thought into coherent “chunks of thought,” (2) Monte-Carlo-based relative outcome labeling for stepwise supervision, and (3) reinforcement learning of a generative judge through GRPO.
The experimental evaluation is extensive and well controlled. Results on ProcessBench and inference-time search clearly demonstrate that the RL-trained generative judge significantly outperforms strong discriminative and SFT baselines. The ablation studies convincingly isolate the impact of each design component, showing that online RL and CoT-based generation both contribute materially to performance.

**Weaknesses:**

While the proposed framework is conceptually strong, several limitations reduce its scientific rigor and generality.
First, all experiments are conducted exclusively on Qwen2.5-based models (1.5B and 7B). These models are known to have been exposed to extensive mathematical corpora during pretraining, leading to potential data contamination and evaluation leakage on benchmarks such as GSM8K, MATH, and ProcessBench. This raises concerns about the true generalization of the proposed method beyond math-specific reasoning or Qwen’s internal distributional biases.

Second, although the authors claim that dense, step-level supervision provides a richer learning signal than outcome-level RLHF, the paper lacks formal theoretical analysis—for example, no proof or derivation is given to quantify the sample-efficiency advantage or convergence behavior of GRPO under stepwise credit assignment. The argument remains mostly empirical.

Third, the evaluation diversity is limited. The framework is only tested on mathematical reasoning tasks; extending it to symbolic reasoning, logic puzzles, or real-world procedural reasoning (e.g., code synthesis or multi-hop QA) would provide stronger evidence of its generality.

Finally, although the generative CoT-based judge improves interpretability, the paper does not analyze the robustness or consistency of the generated rationales—e.g., whether the judge’s reasoning chains are stable under paraphrased inputs, adversarial perturbations, or different sampling temperatures. This limits our understanding of how much of the gain stems from genuine meta-reasoning versus overfitting to specific reasoning patterns.

**Questions:**

1. Data contamination:
Since all experiments use Qwen2.5 models, which are known to contain math-heavy data, how did the authors ensure there is no benchmark leakage (e.g., GSM8K, MATH, ProcessBench)? Would the method still perform well on cleaner model families such as LLaMA or DeepSeek?

2. Generalization:
The paper focuses only on mathematical reasoning. Can the authors provide evidence or discussion on whether STEPWISER generalizes to other reasoning domains, such as logic, commonsense, or code?

3. Robustness:
Are the generative judge’s CoT analyses consistent under paraphrased or slightly perturbed inputs? Have the authors evaluated stability across multiple runs or sampling temperatures?

4. Theoretical grounding:
The paper argues that dense step-level rewards are more effective than sparse ones. Can the authors formalize or provide intuition on why this improves sample efficiency under GRPO?

5. Integration with policy training:
Since STEPWISER can serve both as a training-time reward and inference-time controller, have the authors considered joint optimization with the policy model? What challenges would this introduce?

---

### Official Review · Reviewer_RYiS · 2025-10-30

**Soundness:** 3
**Presentation:** 4
**Contribution:** 3
**Rating:** 4
**Confidence:** 4

**Summary:**

This paper proposes StepWiser, a step-wise GenRM trained by online technique, following SFT. The trained RM can segment steps more reasonably after distillation and score at the step level. The author also introduces a relative improvement approach on step-level estimation. Experiments and analysis have demonstrated the effectiveness of the method. The ablation study also demonstrates that RL contributes significantly to the performance improvement.

However, due to the risk of data pollution in the backbone model used and the lack of experiments on other models, I believe that this paper is not strongly acceptable; we cannot determine whether the improvement in the effectiveness of generative RM is due to reasoning ability by data pollution or the strong judging ability of the model. The ability to stimulate models on already trained data is what RL excels at.

I know it's too much to ask for more model experiments, but the excessive improvement makes me have these concerns: a comprehensive validation for multiple model experiments should have been available when submitting the paper, or it will raise my concern that the score is improved by the model's reasoning ability (and data pollution).

A score of 4 indicates that there are many key issues that need clarification, not the method itself.
Therefore, if this concern cannot be solved, I will tend not to accept this paper and retain my score. On the contrary, if the method is universal and if the author can argue that such a large cost is reasonable, I am willing to recommend it as a poster.

**Strengths:**

1. The paper is well written, and the method is easy to understand.
2. The experiments show significant performance improvements.
3. The ablation shows that the proposed components are valid.

**Weaknesses:**

1. The main experiments are only conducted based on Qwen2.5, which involves data contamination on several datasets.
2. The method is costly (around 5 days on 8*A100 GPUs), and the paper does not provide fair or cost-comparable baselines, like an RL-trained reasoning GenRMs (cost-competitive reasoning GenRMs without process-level reward).
3. The authors do not prove whether the improvement of the model's judging ability is related to data pollution, and whether the model's judging ability strongly depends on its reasoning ability.

**Questions:**

I have listed several major questions in the Summary and Weakness sections. Some additional minor questions are as follows:

1. Is the StepWiser model susceptible to prompt attacking, and how robust is it?
2. If process-level judgment is removed, would training a Reasoning RM with the same computational budget and method yield comparable performance?

For other questions, see weakness.

---

### Official Review · Reviewer_uwtR · 2025-10-31

**Soundness:** 3
**Presentation:** 3
**Contribution:** 2
**Rating:** 4
**Confidence:** 3

**Summary:**

The paper proposes STEPWISER, a generative stepwise judge that evaluates intermediate reasoning steps of LLMs by reasoning about them. Unlike traditional process reward models that treat step evaluation as classification, STEPWISER reframes it as a reasoning problem trained via reinforcement learning (RL). The proposed method comprises three components: (1) a Self-segmentation method, (2) step-wise reward annotation, and (3) online training of judgment reasoning chains. The authors show that STEPWISER can be used to provide rewards at either training or inference time. The authors conducted experiments on ProcessBench and demonstrated improved performance over a classification baseline.

**Strengths:**

- The idea of reframing stepwise evaluation from classification to reasoning is interesting.
- The paper is well written. Each component of their methods is well explained.  Strong ablations clearly isolate the contribution of each component and demonstrate improvements in both training-time reward modelling and inference-time search.
- The results show that their method provides improved inference-time search and better training-time rewards.

**Weaknesses:**

- Despite its framing, the approach does not fundamentally innovate beyond concurrent work on generative process reward models. The “reasoning about reasoning” claim is largely conceptual—since the judge’s reasoning trace is not used in the loss or supervision signal. The RL reward depends solely on the final verdict label.

- All results focus on final-answer correctness. There is no manual evaluation of the judge’s faithfulness or calibration. Annotating a small set of reasoning steps and comparing judge labels to human ratings would have made the claims more convincing.

**Questions:**

- How does segmentation quality and downstream judge performance change when using the same-scale model (e.g., Qwen2.5-1.5B or 7B)?

---

### Note · Authors · 2025-12-10

I have read and agree with the venue's withdrawal policy on behalf of myself and my co-authors.